# A Multi-Strategy Hybrid Sparse Reconstruction Method Based on Spatial–Temporal Sparse Wave Number Analysis for Enhancing Pipe Ultrasonic-Guided Wave Anomaly Imaging

**DOI:** 10.3390/s24165374

**Published:** 2024-08-20

**Authors:** Binghui Tang, Yuemin Wang, Ruqing Gong, Fan Zhou

**Affiliations:** College of Power Engineering, Naval University of Engineering, Wuhan 430030, China; tangbinghui0816@163.com (B.T.); aoi97s@163.com (R.G.); zf422725587@163.com (F.Z.)

**Keywords:** ultrasonic-guided wave, anomaly imaging, sparse wave number analysis, sparse reconstruction

## Abstract

Ultrasonic-guided waves (UGWs) in defective pipes are subject to severe coherent noise caused by imperfect detection conditions, mode conversion, and intrinsic characteristics (dispersion and multiple modes), inducing the limited performance of anomaly imaging. To achieve the high resolution and accuracy of anomaly imaging, a multi-strategy hybrid sparse reconstruction (MHSR) method based on spatial–temporal sparse wavenumber analysis (ST-SWA) is proposed. MHSR leverages the capability of ST-SWA to extract the wavenumber dispersion curves, thereby providing a more refined and precise search space for MHSR. Furthermore, it mitigates the impact of coherent noise by conducting dispersion compensation on the reconstructed signal. The sparse compensated signals through MHSR are employed for sparse reconstruction imaging. To validate the efficacy of the proposed method, UGW testing is performed on the defective steel pipe, and the results demonstrate the significant enhancement of anomaly imaging in defect resolution and positioning accuracy. The lowest estimated errors for axial and circumferential defect positions are 10 mm and 4 mm, respectively.

## 1. Introduction

Pipes, serving as carriers for the transportation of oil, gas, and water, are susceptible to developing defects such as cracks, corrosion, and deformation due to environmental and human factors over their operational lifespan [1,2]. Timely detection of these defects can prevent significant property losses and safety hazards resulting from pipe leaks. Ultrasonic-guided wave (UGW) testing is a novel nondestructive testing technology characterized by long detection distance, full cross-section coverage, minimal attenuation, and high sensitivity; it provides reliable evidence regarding the health status of pipes [3,4,5].

In contrast to bulk waves, UGWs represent mechanical waves that satisfy the boundary conditions of the waveguide, typically manifesting as low-frequency ultrasound below the kHz level [6]. When propagating along the pipe axis, UGWs segregate into axisymmetric and non-axisymmetric modes based on their circumferential energy distribution. The former encompasses longitudinal and torsional modes widely favored for their low signal interpretation complexity [7]. Detection principles for UGWs encompass but are not limited to magnetostrictive [4], electromagnetic [8], piezoelectric [9], and laser techniques [10]. Among these methods, magnetostrictive UGW testing stands out due to its high energy conversion efficiency with few measurement points required at a low cost while being easily implementable, rendering it particularly suitable for ferromagnetic pipes.

Conventional signal processing methods like wavelet transform [11], Chirplet transform [12], and empirical mode decomposition [13] improve the signal-to-noise ratio (SNR) of UGW signals by eliminating non-coherent noise mainly caused by the detection environment, thereby enabling the identification and location of defects using prior knowledge of group velocity at the center frequency. However, due to the imperfect excitation conditions, mode conversion, and the intrinsic characteristics of UGWs (dispersion and multiple modes), coherent noise may inevitably appear in UGW signals. The above approaches fail to address the influence of coherent noise and are limited to estimating the one-dimensional position.

With the aim of eliminating coherent noise, sparse reconstruction methods, as an integral component of compressed sensing, have gradually gained wide attention in UGW testing [14]. The greedy algorithm, exemplified by the matching pursuit (MP) [15], is among the most widely utilized methods, alongside algorithms such as orthogonal matching pursuit (OMP) [16], basis pursuit (BP) [17], and compressive sampling matching pursuit (CoSaMP) [18], which are also employed for achieving sparse reconstruction of UGW signals. By retaining only the sub-signals that capture the characteristics of the waveguide (defects, end-faces), coherent noise can be eliminated. The primary challenge in UGW testing using sparse reconstruction methods lies in the establishment of an efficient and comprehensive dictionary, the level of correspondence between the model for constructing the dictionary and the UGW signal directly influences the precision of the sparse reconstruction. In addition to the classic Gabor [19] and Chirplet [20] models, UGW models have also been developed, such as asymmetric Gaussian Chirplet (AGC) [21] and nonlinear Hanning-windowed Chirplet (NHWC) [22].

With the aim of providing comprehensive spatial information about defects, sensor array-based UGW anomaly imaging can be a feasible approach. Guided wave tomography (GWT), validated extensively since the early 1990s [23,24,25], achieves high defect resolution and defect location accuracy via meticulous mechanical scanning but demands dense sensor deployment due to its sensitivity, thus restricting widespread application owing to bulky expensive scanning equipment. The delay-and-sum (DAS) method is a widely used UGW anomaly imaging method, which involves delaying and shifting each measured signal based on specific spatial coordinates, summing the amplitude of the signals to obtain image intensity at each position, and then visualizing the results to generate defect images [26]. Kim et al. [27] enhanced the resolution of DAS by pre-processing measured signals with matching pursuit (MP), resulting in a 62.1% improvement in location accuracy. The reconstruction algorithm for probabilistic inspection of defects (RAPID) utilizes signal differential coefficients and spatial distribution coefficients to describe defect probability through correlation analysis [28]. Zhang et al. [29] proposed a combined UGW imaging method that integrates RAPID with time-of-flight compensation, demonstrating increased stability and accuracy in defect localization with an average positioning error of only 7 mm. The multiple signal classification (MUSIC) commonly used in radar and sonar fields has also been applied to UGW imaging, leveraging assumptions of independence between signal and noise for decomposing covariance matrix arrays into subspaces through eigenvalue decomposition to estimate arrival direction using orthogonality [30]. Zuo et al. [31] introduced a model-based 2D MUSIC whose accuracy and resolution were validated by experiments.

However, the above conventional UGW anomaly imaging methods heavily depend on the UGW group velocity *c_g_* for defect localization. In case the *c_g_* dispersion curve deviates from the actual scenario, it may lead to a relatively significant error in defect positioning. Considering the relationship between *c_g_* and wavenumber *k*, the above problem is transformed into how to obtain the accurate *k*. Cai et al. [32] proposed a linear dispersion signal reconstruction method that obtains *k* by calculating the phase difference between one-dimensional signals and excitation signals. Although this method is simple, its accuracy is often compromised by the low SNR of the measured signal. Despite UGW’s application in the medical field, several dispersion curve estimation approaches based on the time–space signals can be utilized. Barzegar et al. [33] applied the classical 2D-FFT to obtain the wavenumber distribution of Lamb UGWs, and the comparison of estimated and theoretical dispersion curves showed low error. Zeng et al. [34] recommended the adoption of short-time Chirp–Fourier transform, ridge tracking, and the Vold–Kalman filter to obtain the Lamb wave dispersion curves. Chen et al. [35] extracted the dispersion curves via the rotation invariant technique (ESPRIT), which is inducive to the model-based elastic property estimation.

In order to address the challenges posed by coherent noise and inaccurate dispersion curves, which lead to low imaging resolution and poor defect location accuracy, this paper proposes a novel UGW anomaly imaging method called multi-strategy hybrid sparse reconstruction imaging (MHSRI) based on spatial–temporal sparse wavenumber analysis (ST-SWA). The working principle of MHSRI mainly consists of two steps, namely, pre-reconstructing the measured array signals, and then performing sparse reconstruction imaging based on the reconstructed signal. Step 1 can be subdivided into three sequential stages: (1) ST-SWA is utilized to obtain the wavenumber dispersion curves, which offer more precise dispersion parameters for the subsequent processing; (2) modified orthogonal matching pursuit (MOMP) is utilized to construct the multi-mode dispersive UGW signals; (3) dispersion compensation is utilized to eliminate correlated noise by converting the constructed signal to a single-mode non-dispersion UGW signal. Thanks to the accurate dispersion curves and the sparse signals without coherent noise, the effect of sparse reconstruction imaging can be guaranteed, and thus, the resolution and accuracy of MHSRI can be enhanced.

## 2. Pipe UGW Characteristics

The pipe material used is carbon steel with a density of 7800 kg/m^3^, an elastic modulus of 217 GPa, a Poisson’s ratio of 0.28, an inner radius of 25 mm, and a wall thickness of 5 mm. The wavenumber curves *k*(*f*) for the pipe are solved using the semi-analytical finite element method (SAFE) in the AXISAFE package [36]. Subsequently, the phase velocity *c_p_*(*f*) and group velocity *c_g_*(*f*) are calculated using formulas *c_p_*(*f*) = *f*/*k*(*f*) and *c_g_*(*f*) = d*f*/d*k*(*f*), resulting in the dispersion curves depicted in Figure 1. For clarity, we represent the non-axisymmetric mode F(*n*,*m*) (where *n* is the circumferential order and *m* is the family number) as L(*n*,*m*) and T(*n*,*m*). It can be observed from Figure 1a that various modes of UGWs exhibit distinct cut-off frequencies. Near these cut-off frequencies, there is evident nonlinearity in *k*; however, as frequency increases, *k* gradually tends to increase linearly. This nonlinear behavior in *k* results in varying *c_p_* and *c_g_*—a phenomenon known as dispersion—as illustrated in Figure 1b,c. Furthermore, at any given frequency, multiple different modes may exist concurrently.

## 3. Pipe UGW Propagation Model

In an ideal scenario, the spectrum of the UGW signal can be computed utilizing $Y\left(\omega \right)=Y_{e}\left(\omega \right)e^{-ik\left(\omega \right)d}$, where *Y_e_*(*ω*) is the spectrum of the excitation signal and *d* is the propagation distance. *k*(*ω*) can be directly substituted into the dispersion curve shown in Figure 1. For convenience of operation, this paper uses the third-order Taylor expansion of *k*(*ω*) as an asymptotic solution to construct the UGW propagation model, which is named the nonlinear dispersive model (NDM) in this paper, and its expression in the time domain is as follows [37]:(1)$$y\left(t\right)=IFFT\left\{Y_{e}\left(\omega \right)e^{-i\left[k_{0}+k_{1}\left(\omega -\omega_{c}\right)+k_{2}{\left(\omega -\omega_{c}\right)}^{2}+k_{3}{\left(\omega -\omega_{c}\right)}^{3}\right]d}\right\}$$
where *k*_0_ = *k*(*ω_c_*), *k*_1_ = (d*k*/d*ω*)|*ω_c_*, *k*_2_ = (1/2)(d^2^*k*/d*ω*^2^)|*ω_c_*, and *k_3_* = (1/6)(d^3^*k*/d*ω*^3^)|*ω_c_*. Utilizing a five-cycle modulated sinusoidal signal with *f_c_* = 100 kHz as an excitation, Equation (1) is employed to synthesize the time domain signals (asymptotic signals) of L(0,2) and L(0,1) at *d* = 1 m and *d* = 2 m, which are then compared with the theoretical signals based on the SAFE solutions. In Figure 2a, the level of agreement between the asymptotic and the theoretical signals is remarkably high, particularly for L(0,2), where the two signals exhibit almost identical behavior. In contrast, for L(0,1) characterized by stronger dispersion, only a slight difference is observed between the two signals at the tail of the wave packet.

Subsequently, the phase difference *Φ*(*f*) between the asymptotic and the excitation signal was calculated, and wavenumber and phase velocity were obtained by *k*(*f*) = *Φ*(*f*)/*x* and *c_p_*(*f*) = *f*/*k*(*f*), as illustrated in Figure 2b,c, where the gray gradient area represents the excitation spectrum. Additionally, both time–frequency distribution and instantaneous frequency *IF*(*t*) were derived using the pseudo-Wigner–Ville transform. Furthermore, by converting from the time axis to the group velocity axis using *c_g_* = *x*/*t*, the group velocity dispersion curves were obtained, as shown in Figure 2d,e. It can be inferred that there exists substantial alignment between asymptotic and theoretical solutions across various parameters, including time domain signals, *k*, *c_p_*, and *c_g_*, thus indicating the efficacy of this model as a supersonic UGW propagation model.

## 4. Proposed Methodology

### 4.1. Spatial–Temporal Sparse Wavenumber Analysis

The accuracy of UGW anomaly imaging is largely dependent on *c*_g_, but the actual waveguide’s *c*_g_ may not agree with the theoretical value based on SAFE; therefore, it is crucial to obtain an approximation of *c*_g_. In order to ensure the resolution obtained by the above wavenumber analysis methods, it is necessary to increase sampling in both time and space dimensions as much as possible, which leads to a large number of sensors and massive measurement data, making the operation process somewhat cumbersome. Inspired by the idea of compressed sensing, sparse wave number analysis (SWA) utilizes the sparsity of *k*(*f*) to reconstruct the sparse *k*-*f* distribution by the limited sensors [38].

Suppose the measured signal spectrum matrix is **Y**(*d*,*f*); SWA achieves reconstruction through the dictionary **D**(*d*,*k*) and coefficient matrix **x**(*k*,*f*), i.e., **Y** = **Dx** + **e**, where **e** is the reconstruction error and the atoms in **D** are expressed as g(m,n)=βe−ikndm, where *β* is the normalized coefficient with unit two-norm. Currently, SWA has many variants, such as spatial SWA [39], 2D-SWA [40], and polar SWA [41]. In this paper, a 2D-SWA-like spatial–temporal SWA (ST-SWA) is proposed to obtain a higher-resolution *k*-*f* distribution, which reconstructs the time domain signal matrix **y**(*d*,*t*) using the left and right two dictionaries **D_L_**(*d*,*k*) and **D_R_**(*f*,*t*), respectively, as follows:(2)yM×N=DLM×KxK×QDRQ×N
where the atoms of **D_L_** and **D_R_** are expressed as gL(m,n)=βe−ikndm and gR(m,n)=βe−i2πfmtn, and *M*, *K*, *Q*, and *N* are the number of measurements in the domains of distance, wavenumber, frequency, and time, respectively. The pseudocode of the ST-SWA is shown in Algorithm 1:
**Algorithm 1** ST-SWA**Input:** **y**, **D_L_**, **D_R_**, *τ***Initialize**: R0←y, Λ0←∅, Z0←DLTR0DRT, x←O**Output: x****for***i* = 1: *τ*
**do**Ii←argmax(m,n){Zi−1}, Λi←Λi−1∪{Ii}, Ai←DL(Λi)DR(Λi)xIi←argminx‖y−Aix‖, Ri←Ri−1−AixIi, Zi←DLTRiDRT**end for****return X**

Where τ represents the sparsity, **R** represents the residual, **Z** represents the solution based on the residual, Λ is the set of indices **I** = (m,n) representing the row and column indices of the maximum value of **Z**, and $D_{L}\left(\Lambda_{i}\right)$ and $D_{R}\left(\Lambda_{i}\right)$ represent the atoms selected from the corresponding dictionaries. After obtaining the *k*-*f* distribution based on the ST-SWA, the energy peak ridges are extracted and a third-order polynomial fitting is performed, which can yield an estimated *k*(*f*) similar to the third-order asymptotic solution.

### 4.2. Multi-Strategy Hybrid Sparse Reconstruction

Based on the *k*(*f*) from ST-SWA, the reconstruction of multi-mode dispersive UGW signals can be achieved by the modified OMP (MOMP) method proposed in the previously published paper [37]. Given that the large size dictionary is essential for the high reconstruction accuracy of the sparse reconstruction, MOMP employs the dung beetle optimization (DBO) algorithm [42] instead of the greedy search process in OMP, which can rapidly and effectively identify the globally optimal NDM parameters. The dung beetle position **Para** represents the NDM model parameters, Para=[x,Θ1,Θ2,…,ΘK], Θi=[k0i,k1i,k2i,k3i], where *i* denotes the *i*-th mode and *K* denotes the number of modes for reconstruction. The upper and lower bounds of the dung beetle position can then be expressed as Ub=[dmax,ΘU] and Lb=[dmin,ΘL], where log10ΘU=(1+δ)[log10Θ1,log10Θ2,…,log10ΘK], log10ΘL=(1−δ)[log10Θ1,log10Θ2,…,log10ΘK]. The fitness function can be expressed as
(3)Fitness=norm(y−D^Kx^K)
(4)[D^K,x^K]=OMP(D,y,K)
where **y** is the input signal, D=[gΘ1,gΘ2,…,gΘK], and gΘi is the dispersive atom derived by substituting Θi into NDM (Formula (1)). DBO has been designed by ball-rolling, dancing, breeding, foraging, and stealing processes; they correspond to different dung beetle position update strategies, which are explained in detail in Xue’s research [42]. The pseudo-code for MOMP is shown in Algorithm 2.
**Algorithm 2** MOMP**Input:** *N*, *I*, *τ*, **Ub**, **Lb**Initialize: P^←∅, D^←∅, x^←∅**Output:** y^, P^, D^, x^**for** *k* ← 1 to *τ*
**do**Initialize:Para←[Para1;Para2,…,ParaN]
**while** (*i* ≤ *I*) **do**      **for** *j* ← 1 to *N*
**do**          **if**
*j* == ball-rolling dung beetle **then** δ = rand(1);            **if** δ < 0.9 **then**
$\mathrm{Update}\;Para_{j}(i+1)$ by ball-rolling way;            **else**
$\mathrm{Update}\;Para_{j}(i+1)$ by dancing way;            **end if**            $Para_{j}(i+1)=Bounds\left(Para_{j}(i+1),Ub,Lb\right)$;          **end if**          **if**
*j* == brood ball **then**
            $\mathrm{Update}\;Para_{j}(i+1)$ by breeding way, $Para_{j}(i+1)=Bounds\left(Para_{j}(i+1),Ub,Lb\right)$;          **end if**          **if**
*j* == small dung beetle **then**            $\mathrm{Update}\;Para_{j}(i+1)$ by foraging way, $Para_{j}(i+1)=Bounds\left(Para_{j}(i+1),Ub,Lb\right)$;          **end if**          **if**
*j* == thief **then**
            Update Paraj(i+1) by stealing way, Paraj(i+1)=Bounds(Paraj(i+1),Ub,Lb);          **end if**      **end for**      if Fiteness(Para(i+1))<Fiteness(Para(i)) **then** Update **Para**(i);      **end if**      *I* = *i* + 1;**end while**D^K=NDM(Para(I)), x^K=OMP(D^K,y,K), P^=[P^;Para(I)], D^=[D^,D^K], x^=[x^;x^K], y=y−D^K×x^K**end for**return y^←D^x^P^D^x^


The inputs are the population size *N*, the maximum iteration number *I*, the sparsity *τ*, and the bounds **Ub** and **Lb**. The outputs are the reconstruction signal $\hat{y}$, the global optimal NDM parameter set $\hat{P}$, the dispersive dictionary $\hat{D}$, and the reconstruction coefficient $\hat{x}$. In addition, the bounds’ function is to control the position not to exceed the bounds **Ub** and **Lb**.

It is worth noting that the initial boundaries of MOMP are set based on the theoretical dispersion parameters. Thanks to the estimated dispersion curves by ST-SWA, the search space defined by the upper and lower bounds can be more precisely specified, thereby contributing to enhanced accuracy in reconstruction. To mitigate the impact of coherent noise, the reconstructed signal should be processed by the dispersion compensation, of which the multi-mode dispersive signal can be converted to a signal-mode non-dispersive signal. The output of MOMP also includes the NDM parameter set $\hat{P}$, of which the values of *k_1_* of different modes can be extracted to realize the dispersion compensation. The dispersion compensation can be achieved by the following operation:(5)$$y_{dc}^{\left(0,2\right)}\left(t\right)=\sum_{i=0}^{n}\sum_{j=1}^{m}y_{nd}^{\left(i,j\right)}\left(k_{1}^{\left(0,2\right)}t/k_{1}^{\left(i,j\right)}\right)$$
(6)$$y_{nd}\left(t\right)=IFFT\left\{Y_{e}\left(\omega \right)e^{-ik_{1}\left(\omega -\omega_{c}\right)d}\right\}$$
where *n* and *m* represent the maximum circumferential order and family number, and the subscript (*i*,*j*) corresponds to the order and family number of the given mode. The method that includes ST-SWA, MOMP, and dispersion compensation is called multi-strategy hybrid sparse reconstruction (MHSR), and its flowchart is shown in Figure 3.

### 4.3. Sparse Reconstruction Imaging

In addition to the conventional UGW anomaly imaging methods mentioned in the introduction, sparse reconstruction imaging, inspired by the sparse reconstruction method, represents an effective enhancement in resolution. Xu et al. [43] proposed an anomaly imaging method based on weighted sparse reconstruction, which obtains pixel values at each position in the imaging area by solving the weighted sparse reconstruction problem. For a better understanding of the principles behind sparse reconstruction imaging, its process is illustrated in Figure 4.

Suppose that the number of excitation-receiving pairs used for UGW imaging is L. The hollow cylindrical pipe is unfolded into a planar rectangle, and the imaging area is set and discretized into N nodes. Since each node may be a reflection source, a dictionary **D** can be constructed that contains L × N signals, and the reconstruction coefficient **x** with a specified sparsity τ can be obtained by solving the following problem:(7)$$\underset{x}{min}{\left\|x\right\|}_{0}\;\mathrm{subject}\;\mathrm{to}\;\left\|y-Dx\right\|\leq \sigma$$
where σ is the standard deviation of the error term **e** (**y** = **Dx** + **e**), **y** is the matrix that contains L different measured signals, represented as **y** = [**y**_1_,**y**_2_,…,**y**_L_]^T^, and **D** is constructed in the following way:(8)$$D=\left[\begin{array}{ccc} g_{1,1} & \ldots & g_{1,N}\\ \vdots & \ddots & \vdots \\ g_{L,1} & \cdots & g_{L,N} \end{array}\right]$$

Due to its capability to eliminate coherent noise and reconstruct single-mode non-dispersive signals, the measured signal **y** can undergo preprocessing using MHSR. Therefore, the atom **g**_i,j_ in **D** can be constructed through Equation (6), where *k*_1_ of the desired mode can be extracted from $\hat{P}$ obtained by MHSR, and the propagation distance *d* can be calculated by $\sqrt{{\left(dx_{j}-dx_{i}^{e}\right)}^{2}+{\left(dy_{j}-dy_{i}^{e}\right)}^{2}}+\sqrt{{\left(dx_{j}-dx_{i}^{r}\right)}^{2}+{\left(dy_{j}-dy_{i}^{r}\right)}^{2}}$, where *dx* and *dy* denote the two-dimensional coordinates of the node or sensor, the superscript e and r represent the excitation and receiving sensors, and the subscript i and j represent the serial numbers of signals and nodes.

It is worth noting that the reconstruction coefficient **x** is an N × 1 dimensional matrix; it should be rearranged according to the node distribution, and the absolute value of the rearranged matrix denotes the pixels in the two-dimensional imaging area. Due to the strong sparsity of **x**, with only a small number of non-zero pixels, it is feasible to transform the two-dimensional imaging results into a three-dimensional display for clear visualization of defects.

## 5. Validation of Synthetic UGW Signals

### 5.1. Estimation of Wavenumber Dispersion Curves

The synthetic spatial–temporal signals were constructed using NDM, incorporating L(0,2) and L(0,1) modes. The spatial measurements were set to 50 with a measurement spacing of 0.02 m, and the initial measured position was at 0.5 m. Gaussian white noise was introduced to the synthetic spatial–temporal signal to generate noisy signals with different SNRs, as depicted in Figure 5, where the lower blue signal represents the time domain signal of the final measured position. As the measurement position advanced, the propagation distance of the UGW increased, leading to a gradual diffusion of energy in the time domain for the dispersive L(0,1) mode with weak dispersion, resulting in temporal broadening of the wave packet. Conversely, for L(0,1) with weak dispersion, there is minimal change observed in wave packet width. Additionally, when SNR decreases to 0.46 Db, both L(0,2) and L(0,1) gradually become indistinguishable amidst non-coherent noise.

Using 2D-FFT, VSD, SWA, and ST-SWA to process noisy signals in Figure 5, *k*-*f* distributions are obtained, as shown in Figure 6a–d. The resolution in all four methods is affected by noise, especially for 2D-FFT and VSD, whose peak ridge resolution is seriously affected by noise. Compared with 2D-FFT and VSD, the peak ridge resolution of SWA and ST-SWA is significantly improved, as evidenced by the narrower peak ridge width and lower noise content. Wavenumber dispersion curves are extracted from the peak ridge, and the estimated errors are calculated, as shown in Figure 6e, using SNR = 7.66 dB for L(0,2) as an example. It can be observed that the overall minimum estimated error is achieved by ST-SWA. The average estimated errors of *k*(*f*) in different SNRs are calculated, and the results are shown in Figure 6f–h. The 2D-FFT and VSD have L(0,2) average error within 1.2 to 1.6 m^−1^ and L(0,1) average error within 0.5 to 1.0 m^−1^. The average error of ST-SWA is the lowest, and it shows strong noise resistance, even in the low SNR (0.46 dB) case, with the average error of L(0,2) and L(0,1) being only 0.34 m^−1^ and 0.27 m^−1^, respectively. Therefore, using ST-SWA for extracting *k*(*f*) based on peak ridges is feasible.

### 5.2. Sparse Reconstruction of Dispersive Perturbated Signals

Considering potential deviations between the actual *k*(*f*) and its theoretical value, it is imperative to conduct a dispersion perturbation analysis to validate the efficacy of the MHSR. NDM (Formula (1)) was employed to synthesize multi-mode dispersive UGW signals encompassing L(*n*,*m*) (*n* = 0–2, *m* = 1–2), with an energy ratio of 0.88:0.12, and the propagation distances of *d*_1_ = 1 m and *d*_2_ = 2 m. Assume the dispersion perturbation parameter set is $\log_{10}\Theta_{p}=(1-\delta )[\log_{10}\Theta_{1},\log_{10}\Theta_{2},\ldots ,\log_{10}\Theta_{6}]$, where $\Theta_{i}=[k_{0}^{i},k_{1}^{i},k_{2}^{i},k_{3}^{i}]$, and the perturbated coefficient δ is set to 0.1, 0.05, −0.05, and −0.1, respectively. By introducing random Gaussian noise as non-coherent noise, noisy perturbated signals (NPS1–NPS4) were generated, as illustrated in Figure 7a–d. Utilizing ST-SWA, the *k*(*f*) of NPS1–NPS4 were estimated and are illustrated in Figure 7e–h. The figure demonstrates that the resolution of different modes is correlated with their energy in the time domain. Due to the low signal amplitude of L(*n*,1) in the time domain, which is nearly overwhelmed by non-coherent noise, its peak ridge resolution is notably limited. The *k*(*f*) of L(0,2), exhibiting the highest energy in the time domain and superior resolution, was isolated for extraction and subsequent calculation of its perturbated coefficient. Assuming that other modes share a similar disturbance coefficient as L(0,2), we obtained estimated *k*(*f*) for all modes, as depicted in Figure 7e–h. The results indicate a high consistency between the estimated *k*(*f*) and trends observed in the peak ridges within the *k-f* distribution.

Because the boundaries of MHSR are based on the estimated *k*(*f*) of ST-SWA, δ in $\log_{10}\Theta_{U}=(1+\delta )[\log_{10}\Theta_{1},\log_{10}\Theta_{2},\ldots ,\log_{10}\Theta_{K}]$ and $\log_{10}\Theta_{L}=(1-\delta )[\log_{10}\Theta_{1},\log_{10}\Theta_{2},\ldots ,\log_{10}\Theta_{K}]$ can be adjusted to a reduced value (0.02). The boundaries of MOMP are based on the theoretical *k*(*f*), and δ should be adjusted to a large value (0.1). In addition to the aforementioned variations in parameter settings, both MHSR and MOMP exhibit identical parameter configurations: dmax and dmin of Ub and Lb are 3 m and 0 m, population size *N* is 100, the maximum iteration number *I* is 200, and the sparsity *τ* is 2. Utilizing MHSR and MOMP for processing NPS1–NPS4, Figure 8a–d displays the reconstructed signals and residual signals. It can be observed that both MHSR and MOMP demonstrate the capability to achieve satisfactory reconstruction results in the presence of non-coherent noise. However, compared with MHSR, it can be noticed that the reconstructed signals of MOMP exhibit larger reconstruction errors, of which the reconstruction error ranges of MHSR and MOMP are [0.15, 0.33] and [0.29, 0.52]. We then calculated the residual signals after the sparse reconstruction, as shown in Figure 8e–h. It can be found that the reconstruction accuracy of MHSR remains high even in the presence of dispersion disturbances and non-coherent noise. The overall amplitude of the residual signal is minimal, and the reconstruction performance of the *d_1_* = 1 m wave packets of NPS2 is suboptimal, primarily due to the elevated levels of non-coherent noise. However, the overall amplitude of the residual signals in MOMP is relatively high, particularly for NPS1, where the amplitude of the residual signal for L(*n*,1) in MOMP reaches 0.1, which is attributed to the broader bounds derived from theoretical *k*(*f*).

Because the MHSR reconstructed signals are multi-mode dispersive UGW signals, it is necessary to perform de-dispersion as a preliminary step. Figure 9a–d presents the non-dispersive components of MHSR signals. Utilizing Formula (5), the non-target modes are compensated into the target mode L(0,2) based on the multi-mode non-dispersive signals. To further emphasize the reconstruction accuracies of MHSR and MOMP, the dispersion compensated signals of MHSR and MOMP are illustrated in Figure 9e–h, and the theoretical dispersion compensated signals are also depicted. The discrepancy in reconstruction errors between MHSR and MOMP is exemplified with respect to the dispersion compensated signals. The MHSR compensated signals closely align with the theoretical signals, exhibiting only minor discrepancies in amplitude. In contrast, the MOMP compensated signals not only display significant deviations in amplitude from the theoretical signals but also exhibit substantial disparities in arrival times.

The reconstruction accuracies of the dispersion compensated signals are largely contingent upon the precision of the reconstructed dispersion parameters, with particular emphasis on the influence of the reconstructed propagation distance *d*. The reconstructed propagation distance errors of MHSR and MOMP are calculated, which are shown in Table 1. In comparison to MOMP, the search space for the dispersion parameters in MHSR is more compact and precise, resulting in a significantly reduced reconstruction error for the propagation distance. Particularly, for NPS3, the errors *d*_1_ and *d*_2_ are merely 0.96 mm and 1.44 mm. However, the overall reconstruction error level for MOMP is relatively high, particularly with respect to NPS2, with *d*_1_ and *d*_2_ reaching 9.81 mm and 28.31 mm.

### 5.3. Pipe Sparse Reconstruction Imaging

Considering that the excitation of L(0,2) needs to be achieved through the form of circumferential full coverage, only one excitation sensor is needed. Eight receiving sensors R_1_–R_8_ (π/16–15π/16) are evenly distributed along the circumference, as shown in Figure 10a. For the convenience of displaying the imaging results, the cylindrical coordinate system of the pipe is transformed from (*r*, *θ*, *z*) to the cartesian coordinate system (*z*, *r_θ_*). Assuming that the defect is located at the axial and circumferential positions of *d_z_* and *d_θ_*, the propagation distance of UGW from the excitation to the defect to R_i_ can be calculated by $d_{z}+\sqrt{{\left(d_{z}-d_{z}^{i}\right)}^{2}+{\left(d_{\theta }-d_{\theta }^{i}\right)}^{2}}$.

The pipe’s outer radius is 60 mm and the wall thickness is 5 mm. The pipe is unfolded along the mid-section of R_1_ and R_8_, the axial coordinates of R_1_–R_8_ are 30 mm, and the plane coordinates of defects D_1_ and D_2_ are (1000 mm, 100 mm) and (2000 mm, 200 mm), respectively. Although the pipe can be unfolded into a plane for imaging, the difference between the pipe and plate is that the UGW propagation in the unfolded plane of the pipe has periodicity, i.e., the reflected UGW signal will arrive at the receiving sensor along the shortest path as shown in Figure 10b. For the convenience of analysis, we take the unfolded plane Ω_0_ and its adjacent planes Ω_1_ and Ω_−1_ for explanation. After the UGW reflects at the defect near Ω_1_, it will first arrive at the receiving sensor in Ω_1_ instead of Ω_0_ along the shortest path.

The received signals of R_1_–R_8_ were synthesized with the perturbation coefficient δ set to 0.1, SNR = 10 dB, and the signal amplitude inversely proportional to the propagation distance, and the results are depicted in Figure 10c. The circumferential distribution of the defect echo’s amplitude in the received signal is computed and illustrated in Figure 10d. It can be observed that the circumferential position of the defect can be approximately determined if the resolution of the defect is high. The initial estimated circumferential positions of D_1_ and D_2_ lie roughly between 3π/16–5π/16 and 7π/16–9π/16, respectively.

To distinguish them, the sparse reconstruction images based on MHSR and MOMP are denoted as MHSRI and MOMPI, respectively. We compared the proposed methods with conventional imaging methods, among which the expressions of the conventional imaging methods are as follows:(1)Delay and sum (DAS) [26]
(9)$$P_{x,y}^{DAS}={\left|\sum_{i=1}^{L}y_{i}\left(t+\frac{d_{ixy}}{c_{gc}}\right)\right|}^{2}$$
where **y**_i_ is the measured signal which is delayed *d_ixy_*/*c_gc_* in the *i*-th wave path, *d_ixy_* is the sum of the distances from the actuator to the point (*x*, *y*) and from the point (*x*, *y*) to the sensor, and *c_gc_* is the group velocity at the central frequency.

(2)Eigenvalue decomposition (EVD) [44]

Similar to DAS, prior to performing EVD imaging, delayed processing of the measured signal is necessary. The delayed signal is designated as $\hat{y}$. The covariance matrix **C** of the delayed measured signal is resolved, and its eigenvalue decomposition is carried out as follows.
(10)$$C=U\Sigma U^{*}=\mathrm{diag}\left(\lambda +\sigma^{2},\sigma^{2},\ldots ,\sigma^{2}\right)$$
where **U** is the matrix of eigenvectors, **Σ** is the eigenvalues corresponding to the main diagonal, λ is the eigenvalue of the defect scattering signal, and σ is the noise power. Based on the above eigenvalue decomposition, the largest eigenvalue is selected as the pixel value of the node (*x*, *y*).
(11)$$P_{x,y}^{EVD}=\max\left[diag\left(\Sigma \right)\right]$$

(3)Reconstruction algorithm for probabilistic inspection of damage (RAPID) [28]

In RAPID, the baseline signal is needed to construct the signal difference coefficient (SDC). The expression of SDC is as follows:(12)$$SDC=1-\frac{C_{db}}{\sigma_{d}\sigma_{b}}$$
where *C_db_* is the covariance of the detection signal and baseline signal, and *σ_d_* and *σ_b_* are standard deviations of the detection signal and baseline signal. The pixel value of the node (*x*, *y*) is expressed as
(13)$$P_{x,y}^{RAPID}=\sum_{i=1}^{L}SDC_{i}\times E_{i}\left(x,y\right)\times p_{i}\left(x,y\right)$$
(14)$$E_{i}\left(x,y\right)=\frac{\beta -R_{i}\left(x,y\right)}{\beta -1}$$
(15)$$R_{i}\left(x,y\right)=\left\{ \begin{array}{ll} \frac{\sqrt{{\left(x-x_{ri}\right)}^{2}+{\left(y-y_{ri}\right)}^{2}}+\sqrt{{\left(x_{ei}-x\right)}^{2}+{\left(y_{ei}-y\right)}^{2}}}{\sqrt{{\left(x_{ei}-x_{ri}\right)}^{2}+{\left(y_{ei}-y_{ri}\right)}^{2}}} & ,R_{i}\left(x,y\right)\leq \beta \\ \beta & ,R_{i}\left(x,y\right)>\beta \end{array}\right.$$
(16)$$p_{i}\left(x,y\right)=\left\{ \begin{array}{ll} 1-\left|\frac{t_{i}\left(x,y\right)-t_{i}^{tof}}{t_{d}}\right| & ,\left|t_{i}\left(x,y\right)-t_{i}^{tof}\right|\leq t_{d}\\ 0\; & ,\left|t_{i}\left(x,y\right)-t_{i}^{tof}\right|>t_{d}\; \end{array}\right.$$
where *β* is the scaling parameter and Ri(*x*, *y*) is the distance ratio between the indirect path and direct path linking the excitation and receiving sensors, which is given in Formula (13). In the *i*-th wave path, the locations of excitation and receiving sensors are (*x_ei_*, *y_ei_*) and (*x_ri_*, *y_ri_*). The probability *p_i_*(*x*, *y*) is controlled by *t_i_*(*x*, *y*), *t^tof^*, and *t_d_*, where *t_i_*(*x*, *y*) denotes the sum of the actuator and receiver arrival times for the point (*x*, *y*) to the *i*-th path, $t_{i}\left(x,y\right)=\sqrt{{\left(x-x_{ri}\right)}^{2}+{\left(y-y_{ri}\right)}^{2}}+\sqrt{{\left(x_{ei}-x\right)}^{2}+{\left(y_{ei}-y\right)}^{2}}/c_{gc}$, *t^tof^* is the arrival time of defect signal, and *t_d_* denotes the defined TOF error.

(4)Multiple signal classification (MUSIC) [31]

The fundamental concept of the MUSIC algorithm, which is predicated on the uncorrelation between signal and noise, lies in decomposing the covariance matrix of array signals into the signal subspace and the noise subspace through conducting eigenvalue decomposition. Distinct from the aforementioned method, the positioning parameters of the MUSIC algorithm are (*r*, *θ*), and the conversion formulas between them and (x, y) are *x* = *r*cos*θ*, *y* = *r*sin*θ*. The covariance matrix **C** can be calculated by Formula (8), and it can be further decomposed as
(17)$$C=U_{S}\Sigma_{S}U_{S}^{*}+U_{N}\Sigma_{N}U_{N}^{*}$$
where the subscripts S and N represent the signal and noise subspaces, and **U** and **Σ** are the eigenvectors and eigenvalues. **Σ** = diag[λ_1_, λ_2_,…, λ_L_], and the eigenvalues can be arranged in descending order, λ_1_ ≥ λ_2_ ≥ … ≥ λ_k_ ≥ λ_k+1_ = …λ_L_, where k is the number of the reflection source. So, **U**_S_ and **U**_N_ can be expressed as **U**_S_ = [**e**_1_,**e**_2_,…,**e**_k_] and **U**_N_ = [**e**_k+1_,**e**_k+2_,…,**e**_L_], and the pixel value of the node (*r*, *θ*) can be calculated by:(18)$$P_{r,\theta }^{MUSIC}=\frac{1}{A{\left(r,\theta \right)}^{*}U_{N}U_{N}^{*}A\left(r,\theta \right)}$$
where **A**(*r*, *θ*) = [a_1_(*r*, *θ*), a_2_(*r*, *θ*),..., a_L_(*r*, *θ*)]^T^, and the expression of a_i_(*r*, *θ*) is as follows:(19)$$a_{i}\left(r,\theta \right)=\exp\left(\frac{-j2\pi f_{c}}{c_{gc}}\left(\sqrt{r^{2}+{\left(i-m\right)}^{2}d^{2}-2R\left(i-m\right)d\sin\theta }-r\right)\right)$$
where *m* is the serial number of the referential receiving sensor and d is the circumferential spacing of the receiving sensors.

The anomaly imaging results of DAS, eigenvalue decomposition (EVD), RAPID, MUSIC, MOMPI, and MHSRI are presented in Figure 11, where triangles denote actual defect positions. It is worth noting that in RAPID, *β* is adopted as 2 × 10^3^, *t_d_* is defined as 1 × 10^−4^, the baseline signal is the synthetic signal with a propagation distance of 5 m, and the detection signal is the superimposition of the baseline signal and the signals in Figure 10c. We identified the node with the highest pixel value as the defect, extracted the axial and circumferential coordinates of this node, and calculated the estimated defect positioning errors of the above methods by subtracting the actual coordinates, of which the results are presented in Table 2. It is important to note that, due to the presence of two defects in the imaging results depicted in Figure 11, the imaging area should be partitioned into two segments for separate computation of defect location errors.

Due to the perturbation coefficient δ of 0.1 in the synthesized measurement signals, DAS, EVD, RAPID, and MUSIC, based on the theoretical dispersion curve, exhibit diminished accuracy in defect location. Particularly, the axial defect errors of the above methods are very large, varying from 111.5 mm to 180.5 mm. The circumferential defect error is only 1 mm for MUSIC, which is deemed satisfactory, whereas those of other methods range from 23 mm to 34 mm. The limitation in achieving high defect location accuracy using the above methods stems from the substantial disparity between the theoretical dispersion curve and its actual value, as well as the constrained number of sensors which hinders enhancements in location accuracy. In addition, as the measured signal contains both coherent and incoherent noise, the imaging method described above exhibits limited defect resolution. Therefore, threshold processing is essential to achieve satisfactory imaging results.

Compared to conventional UGW imaging methods, both MOMP and MHSR perform a search and optimization of dispersion parameters, resulting in dispersion parameter values that closely approximate the actual values. Consequently, the positioning accuracies of MOMPI and MHSRI are significantly enhanced. In particular, MHSR, which has pre-calculated dispersion curves using ST-SWA, possesses a more constrained and precise dispersion parameter search space compared to MOMP. Therefore, MHSRI shows a reduced defect location error of only (3 mm, 6.5 mm). Furthermore, as the input to MHSRI consists of single-mode non-dispersive signals with dispersion compensation, both coherent and non-coherent noise are effectively eliminated. Consequently, only a limited number of pixel values exhibit non-zero in MHSRI imaging, indicating exceptionally high defect resolution.

## 6. Validation of Experimental UGW Signals

The magnetostrictive UGW testing of pipe was conducted by a proprietary UGW platform, as illustrated in Figure 12a. To mitigate signal interpretability degradation resulting from bidirectional propagation of UGWs, the excitation coil and receiving coil were positioned at the same pipe end. Additionally, the pre-magnetized magnetostrictive patch (FeCo alloy) was inserted between the coil and the pipe to enhance detection sensitivity. The excitation comprised a five-cycle Hann window modulated sine wave with a voltage of 1.5 V and a frequency of 100 kHz, featuring an excitation number of 120 times, an interval of 200 ms, and a sampling frequency of 2 MHz over a sampling time of 6 ms. Two steel pipes (pristine and defective) with identical dimensions—outer radius r_o_: 57.5 mm, wall thickness: 5 mm, length: 6.03 m—were utilized.

We derived *k*(*f*) from the pristine pipe by maintaining the excitation coil consistently at one end (d_z_ = 0 mm) while setting the initial position of the receiving coil at d_z_ = 50 mm, and the acquisition of spatial–temporal signals was achieved by moving the receiving coil with directional movement depicted in Figure 12b, of which the detection spacing and detection times are 2 cm and 50. The spatial–temporal signals obtained are presented in Figure 12c. Subsequently, we employed ST-SWA to yield the *k-f* distribution displayed in Figure 12d. A comparison between extracted *k*(*f*) and theoretical solution revealed minimal disparity, as depicted in Figure 12e, with a perturbation coefficient δ of approximately −0.02.

Eight copper wire coils were wound as receiving sensors and evenly arranged along the pipe circumference, as depicted in the upper left of Figure 13. Two circular defects with a radius of 10 mm and a depth of 3 mm were intentionally introduced on the pipe. Their coordinates in the cylindrical coordinate system are (r_o_, π/6, 2500 mm) and (r_o_, π, 4500 mm), while their coordinates in the plane of the unfolded plane are (2500 mm, 30 mm) and (4500 mm, 180 mm). UGW testing was conducted on the defective pipe to capture the measured signals R1–R8 shown in the lower left of Figure 13. It is evident that apart from the incident wave, defect echo, and pipe end echo signals present in the experimental signals, there exists significant noise that complicates defect identification. Given that defects are the most critical feature of the pipe, we propose preprocessing the experimental signals using variational mode decomposition (VMD) to reduce non-coherent noise and obtain signals around the center frequency [45]. Additionally, the time window function can be applied to obtain the signals that only contain defect echoes, as illustrated in the lower right of Figure 13.

The VMD defect echo signals were processed by DAS, EVD, RAPID, MUSIC, MOMPI, and MHSRI, and the anomaly imaging results are depicted in Figure 14, with triangles representing the actual defect positions. Subsequently, the estimated defect positions obtained from these imaging methods were extracted and the errors between the estimated and actual positions were calculated in Table 3. In comparison to the synthesized noisy disturbed signal, it was observed that due to imperfect experimental conditions, the experimental signal contained higher noise content, which led to reduced resolution of defects in DAS, EVD, and MUSIC imaging. Furthermore, *k*(*f*) extracted by ST-SWA analysis revealed a relatively close resemblance between actual and theoretical frequency dispersion characteristics of the pipeline, indicating that conventional imaging methods yielded relatively accurate estimations of defects compared with those of dispersion disturbance in Section 5.3. However, the enhancement effect is constrained, and the defect positions obtained exhibit axial and circumferential error ranges of [21 mm, 115 mm] and [51 mm, 111 mm], respectively.

Due to the sparse reconstruction capability of MOMP and MHSR for multi-mode dispersion measured signals, along with the ability of dispersion compensation to convert the reconstructed signal into a single-mode non-dispersive signal, coherent noise effects can be completely eliminated. Consequently, MOMPI and MHSRI exhibit high defect resolution. However, as discussed in Section 5.3, due to its reliance on the theoretical frequency *k*(*f*) for reconstruction, MOMP exhibits relatively large reconstruction errors for propagation distance *d*. As a result, the positioning accuracy of MOMPI for defects is not optimal, with an average positioning error of (20 mm, 15 mm). In contrast to MOMP, MHSR utilizes ST-SWA to estimate the dispersion curve and obtain *k*(*f*) closer to actual values than theoretical solutions. This contributes to more accurate reconstruction results for MHSR and lower error in propagation distance *d*. Consequently, the positioning accuracy of MHSRI for defects is enhanced with an average positioning error of (11.5 mm, 11 mm).

## 7. Conclusions

In this paper, a multi-strategy hybrid sparse reconstruction (MHSR) method based on spatial–temporal sparse wavenumber analysis (ST-SWA) is proposed to enhance anomaly imaging for pipe ultrasonic-guided wave (UGW). Compared with conventional anomaly imaging methods, the proposed method exhibits the ability to realize the high resolution and accuracy of anomaly imaging by eliminating the severe influence of coherent noise with the simplified sensor array.

MHSR utilizes ST-SWA to obtain the wavenumber–frequency distribution, of which the wavenumber dispersion curves can be easily estimated by the extraction of energy peak ridges. Based on the estimated dispersion curves and dispersion compensation theory, MHSR achieves the accurate reconstruction of non-dispersive UGW signals in the desired mode, thereby contributing to the outstanding performance of sparse reconstruction imaging. The proposed method was verified by experimental UGW signals, and the results show that the estimated defect location error is very low, with a minimum estimated error of 10 mm in the axial direction and 4 mm in the circumferential direction.

## Figures and Tables

**Figure 1 sensors-24-05374-f001:**
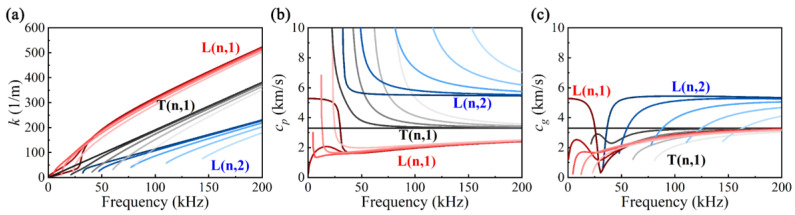
Dispersion curves of pipe UGWs: (**a**) wavenumber *k*, (**b**) phase velocity *c_p_*, (**c**) group velocity *c_g_* (*n* = 0–4).

**Figure 2 sensors-24-05374-f002:**
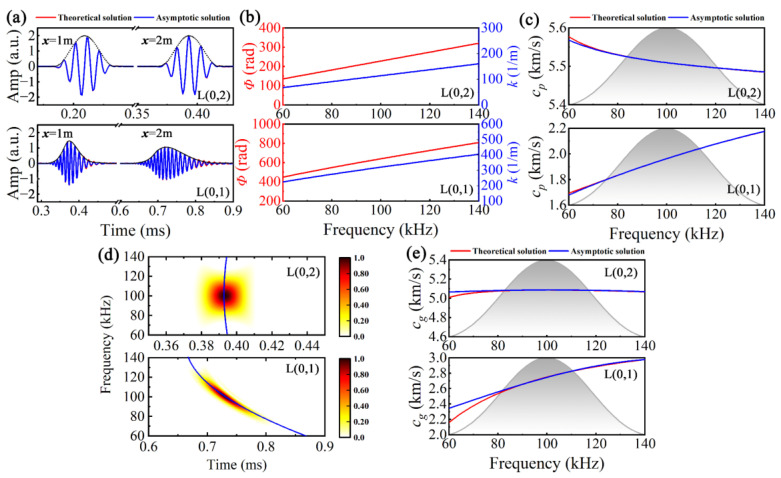
(**a**) Asymptotic signals, (**b**) wavenumber *k*, (**c**) phase velocity *c_p_*, (**d**) time–frequency domain distribution, and (**e**) group velocity *c_g_* of L(0,2) and L(0,1) by NDM.

**Figure 3 sensors-24-05374-f003:**
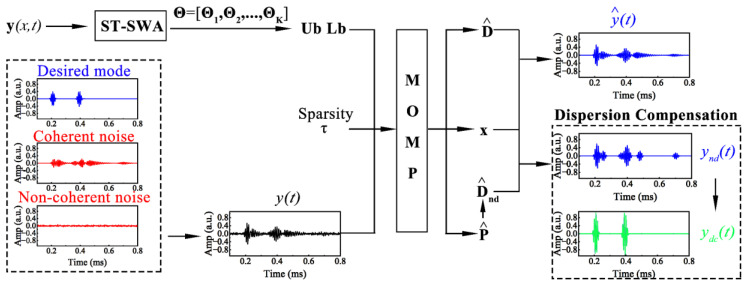
Flowchart of multi-strategy hybrid sparse reconstruction.

**Figure 4 sensors-24-05374-f004:**
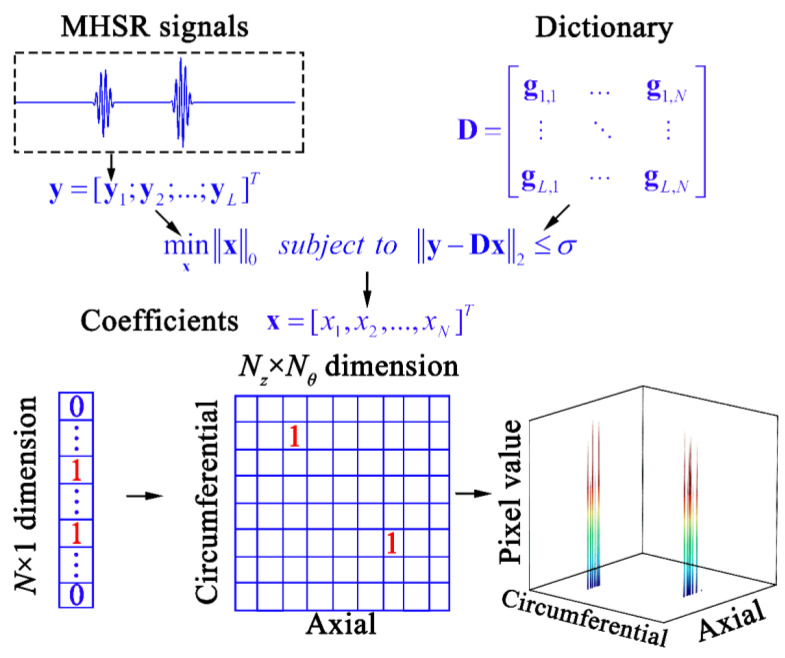
Flowchart of multi-strategy hybrid sparse reconstruction imaging.

**Figure 5 sensors-24-05374-f005:**
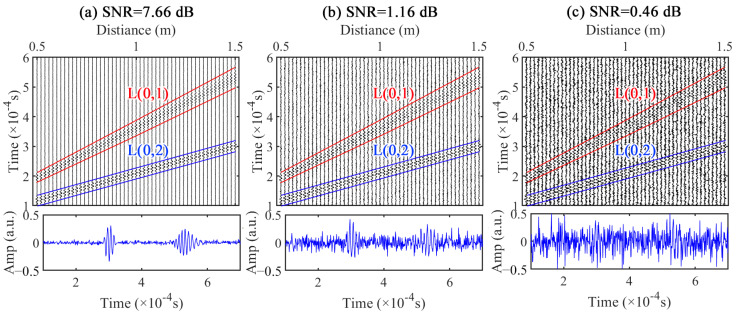
Noisy synthetic spatial–temporal signals: (**a**) SNR = 7.66 dB, (**b**) SNR = 1.16 dB, and (**c**) SNR = 0.46 dB.

**Figure 6 sensors-24-05374-f006:**
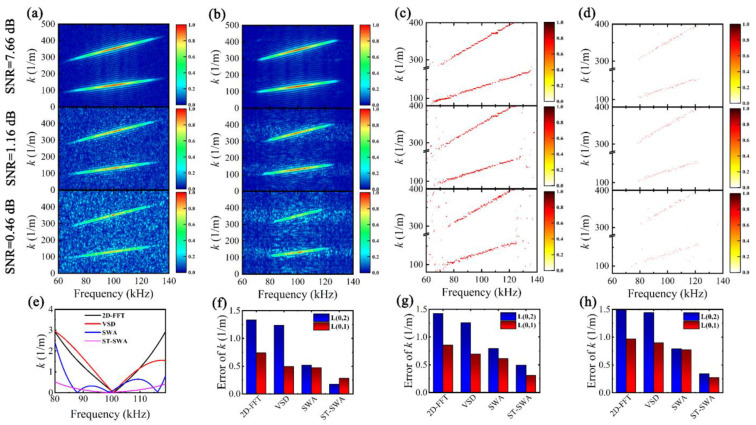
*k*-*f* distribution of noisy time–space signals: (**a**) 2D-FFT, (**b**) VSD, (**c**) SWA, and (**d**) ST-SWA; (**e**) estimated error of *k(f)* at L(0,2) for SNR = 7.66 dB, and the average estimated error of *k(f)* at different SNRs: (**f**) SNR = 7.66 dB, (**g**) SNR = 1.16 dB, and (**h**) SNR = 0.46 dB.

**Figure 7 sensors-24-05374-f007:**
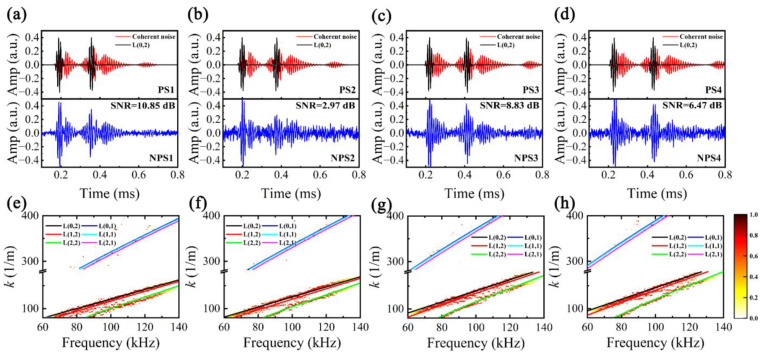
Noisy perturbated signals in time domain: (**a**) NPS1, (**b**) NPS2, (**c**) NPS3, and (**d**) NPS4; *k*-*f* distribution and estimated *k*(*f*) by ST-SWA: (**e**) NPS1, (**f**) NPS2, (**g**) NPS3, and (**h**) NPS4.

**Figure 8 sensors-24-05374-f008:**
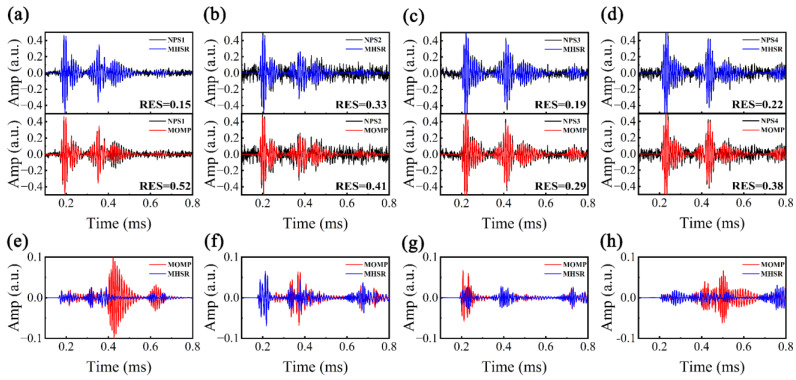
Reconstructed signals and non-dispersive components by MHSR: (**a**) NPS1, (**b**) NPS2, (**c**) NPS3, and (**d**) NPS4; dispersion compensated signals by MHSR and MOMP: (**e**) NPS1, (**f**) NPS2, (**g**) NPS3, and (**h**) NPS4.

**Figure 9 sensors-24-05374-f009:**
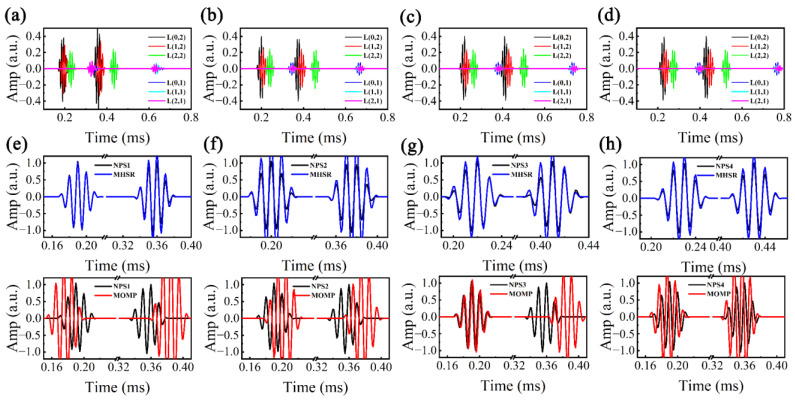
Non-dispersive components of MHSR signals: (**a**) NPS1, (**b**) NPS2, (**c**) NPS3, and (**d**) NPS4; dispersion compensated signals by MHSR and MOMP: (**e**) NPS1, (**f**) NPS2, (**g**) NPS3, and (**h**) NPS4.

**Figure 10 sensors-24-05374-f010:**
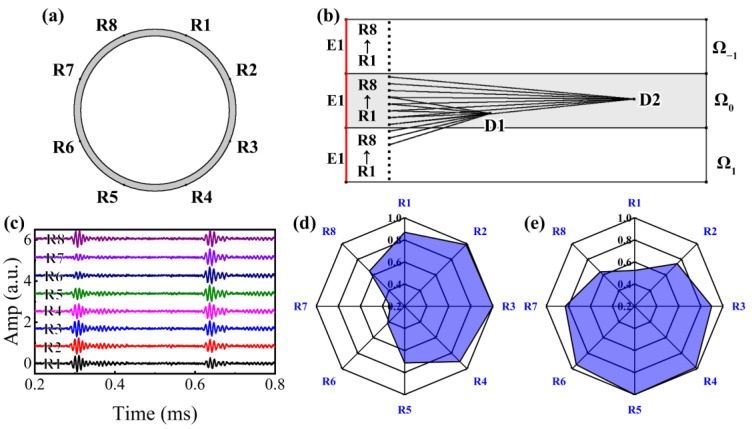
(**a**) Configuration of the receiving sensors, (**b**) propagation paths of the reflected UGW on the unfolded pipe plane, (**c**) synthetic received signals, circumferential distribution of defect echo amplitude: (**d**) D1 and (**e**) D2.

**Figure 11 sensors-24-05374-f011:**
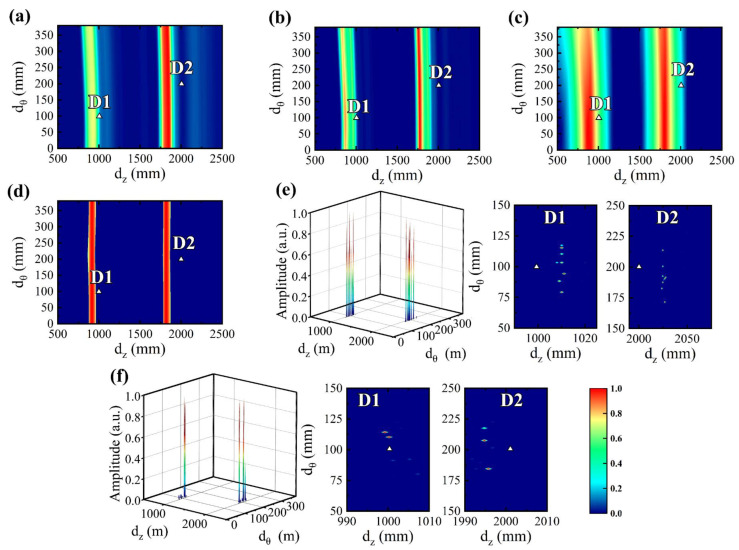
Imaging results of synthetic signals using (**a**) DAS, (**b**) EVD, (**c**) RAPID, (**d**) MUSIC, (**e**) MOMPI, and (**f**) MHSRI.

**Figure 12 sensors-24-05374-f012:**
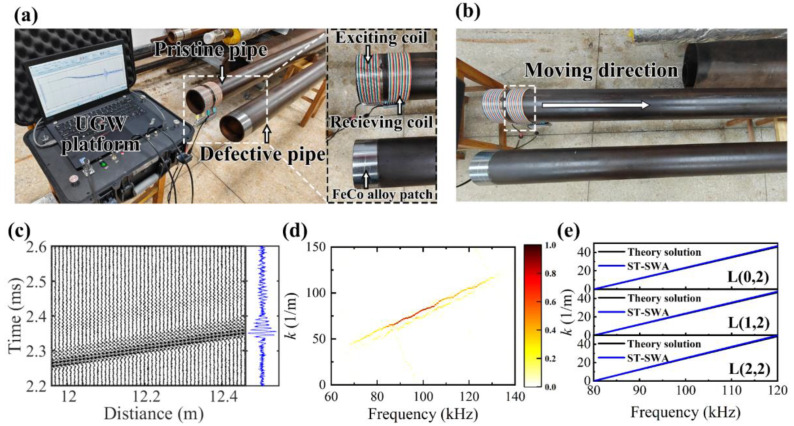
(**a**) UGW testing experiment setup, (**b**) axial detection direction, (**c**) spatial–temporal signal of the pristine pipe, (**d**) *k-f* distribution obtained by ST-SWA, (**e**) *k*(*f*) extracted by ST-SWA.

**Figure 13 sensors-24-05374-f013:**
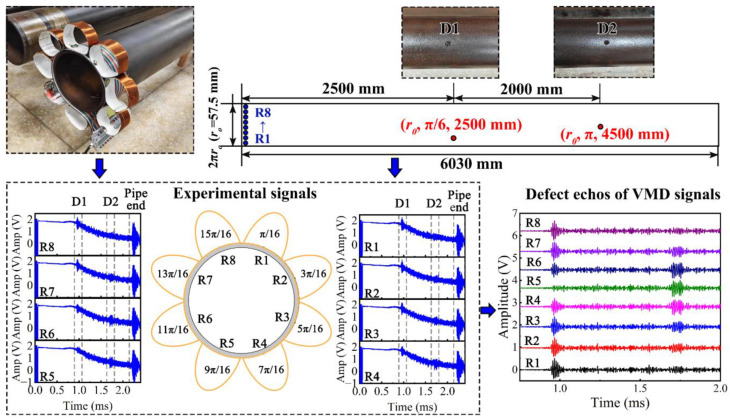
Configuration of the receiving sensors, defect distribution, and measured experimental signals.

**Figure 14 sensors-24-05374-f014:**
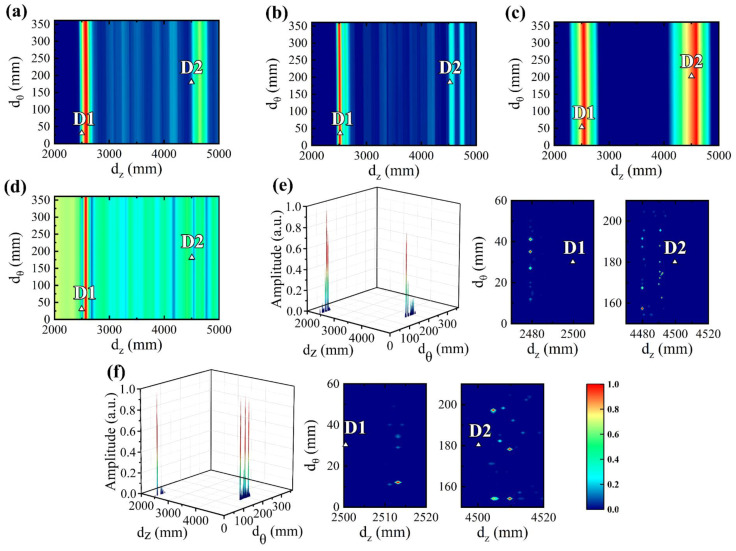
Imaging results of experimental signals using (**a**) DAS, (**b**) EVD, (**c**) RAPID, (**d**) MUSIC, (**e**) MOMPI, and (**f**) MHSRI.

**Table 1 sensors-24-05374-t001:** Reconstructed propagation distance errors of MHSR and MOMP.

	NPS1		NPS2		NPS3		NPS4	
	*d*_1_ (mm)	*d*_2_ (mm)	*d*_1_ (mm)	*d*_2_ (mm)	*d*_1_ (mm)	*d*_2_ (mm)	*d*_1_ (mm)	*d*_2_ (mm)
MHSR	4.24	11.79	4.26	7.05	0.96	1.44	5.15	8.63
MOMP	10.82	24.48	9.81	28.31	3.63	17.28	6.29	16.73

**Table 2 sensors-24-05374-t002:** Estimated defect position errors of synthetic signals by imaging methods (mm).

Error	DAS	EVD	RAPID	MUSIC	MOMPI	MHSRI
D1	(66, 6)	(132, 0)	(130, 11)	(85, 0)	(22, 6)	(1, 5)
D2	(157, 40)	(229, 68)	(182, 45)	(183, 2)	(29, 11)	(5, 7)
Average	(111.5, 23)	(180.5, 34)	(156, 28)	(134, 1)	(25.5, 8.5)	(3, 6.5)

**Table 3 sensors-24-05374-t003:** Estimated defect position errors of experimental signals by imaging methods (mm).

Error	DAS	EVD	RAPID	MUSIC	MOMPI	MHSRI
D1	(76, 92)	(4, 91)	(38, 27)	(70, 86)	(21, 6)	(13, 16)
D2	(153, 10)	(38, 131)	(97, 125)	(83, 24)	(19, 24)	(10, 4)
Average	(114.5, 51)	(21, 111)	(67.5, 101)	(76.5, 55)	(20, 15)	(11.5, 11)

## Data Availability

Data are contained within this article.

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
