# Peer review of "A Multi-Strategy Hybrid Sparse Reconstruction Method Based on Spatial–Temporal Sparse Wave Number Analysis for Enhancing Pipe Ultrasonic-Guided Wave Anomaly Imaging"

_sensors, 2024, doi:10.3390/s24165374_

Round 1

Reviewer 1 Report

Comments and Suggestions for Authors

The paper describes a novel approach to enhancing pipe ultrasonic guided wave (UGW) imaging using a multi-strategy hybrid sparse reconstruction method. The developed and presented methodology has the potential to improve imaging and tomography in pipes using UGW. The authors developed a multi-strategy sparse reconstruction approach to calculate the UGW wavenumber which according to the presented results is capable of accurately reconstructing non-dispersive UGW modes by mitigating the impact of coherent noise. Overall, the paper is well-written with a comprehensive introduction, good methodology section and good presentation of supporting results and findings.

Comments:

1)        Figures are generally small and busy making it hard to read and understand. Please consider increasing the font/figure size or rearranging them.

2)        Some assumptions could be stated more clearly in the methodology section, with sufficient reasoning.

3)        In the presented approach a reflection from potential anomaly in the pipe is synthesized using a complex exponential, Taylor series and added noise. How does this correlate with a reflection from an actual anomaly. How does the size (circumferential, longitudinal, through thickness extent) of the anomaly affect your approach? A discussion about this could help the paper in my opinion.

4)        Ln 35: The value of frequency is missing

5)        Ln 117: “In an ideal scenario” this is vague. Please provide explanation/context. Are the authors referring to no attenuation? Also, since this work is related to pipe, the authors might want to consider talking about the presence of fluid(pressure) internally, and how does it affect their approach.

6)        Ln 127: “The level of agreement…” I do not see a reference to a figure. Is this talking about Fig2a?

7)        Ln 182: “NDE” or “NDM”?

8)        Ln 224-226: Please correct these lines.

9)        Ln 230,232: Hard to follow the comparison made for fig.5 using horizontal and vertical.

10)   fig.5: Has a lot of information that is not labeled. For example, fig5(a-d), are these the k-f curves for all 3 SNR. If yes, please annotate appropriately. Also, there are only two color bars for three rows of plots. (see comment 1)

11)   Ln 254: “NDE” or “NDM”?

12)   Fig6: Please see comment 1

13)   Table 1 & Table 2: Could you please be more specific on how the error is calculated? Axial error seems intuitive but how do the authors define circumferential error? For example, in fig.9a, I see two bands where the max is uniform along d­ϑ for each of the D1 and D2.

Author Response

The complete version of the reply can be downloaded as an attachment.

Dear Editors and Reviewers:

Thank you for your letter and the comments concerning our manuscript entitled “A multi-strategy hybrid sparse reconstruction method based on spatial-temporal sparse wave number analysis for enhancing pipe ultrasonic guided wave anomaly imaging” (sensors-3117990). Those comments are all valuable and very helpful for revising and improving our paper, as well as the important guiding significance to our researches. We have studied comments carefully and have made correction which we hope meet with approval. Revised portion are marked in yellow in the paper. The corrections in the paper and the responds to the comments are as follows:

Q1: Figures are generally small and busy making it hard to read and understand. Please consider increasing the font/figure size or rearranging them.

A1: Thank you for your comments. According to your suggestion, I have magnified the full-text images to an appropriate degree in order to ensure clear display of the contained details.

Q2: Some assumptions could be stated more clearly in the methodology section, with sufficient reasoning.

A2: Thank you for your comments. Similar comments are also provided by Reviewer 3, prompting me to revise and reorganize the theoretical explanations of the algorithms in Section 4 to ensure comprehensive elucidation of all variables and symbols.

Q3: In the presented approach a reflection from potential anomaly in the pipe is synthesized using a complex exponential, Taylor series and added noise. How does this correlate with a reflection from an actual anomaly. How does the size (circumferential, longitudinal, through thickness extent) of the anomaly affect your approach? A discussion about this could help the paper in my opinion.

A3: Thank you for your comments. Before answering your question, I would like to explain how the MHSR system works. The working principle of MHSR mainly includes three steps.

The first step is to use the ST-SWA to obtain an estimated dispersion curve of the actual ultrasonic guided wave, compared with the theoretical solution, the estimated solution is closer to the actual solution;

The second step is to perform a third-order Taylor expansion on the estimated dispersion curve to obtain dispersion parameters, based on the estimated dispersion parameters, MOMP can achieve sparse reconstruction of multi-mode dispersive ultrasonic guided wave signals with lower reconstruction accuracy;

The third step is that since the reconstructed signal is the superposition of multi-mode dispersive ultrasonic guided wave signals, the output of MOMP includes not only the reconstructed signal but also the dictionary used for reconstruction and the reconstruction coefficient x. Using the output results, the non-dispersion and dispersion compensation are performed sequentially, and the high signal-to-noise ratio of the single-mode non-dispersion reconstructed signal obtained finally is very high, which is beneficial to improving the resolution and positioning accuracy of defects in ultrasonic guided wave imaging.

Defect signals caused by actual anomalies are essentially multi-mode dispersive ultrasonic guided wave signals. Based on the above analysis, MHSR can reconstruct multi-mode dispersive ultrasonic guided wave signals and convert them into single-mode non-dispersive signals through dispersion compensation, which can eliminate coherent noise and improve the signal-to-noise ratio.

Therefore, MHSR is only one of the sparse reconstruction methods for processing the actual anomaly signals, and the effect of anomaly size (circumferential, longitudinal, through thickness extent) on the reflection signal may not be a significant factor affecting the accuracy of the MHSR reconstruction. Of course, your suggestions have provided me with a new direction for my future research, and I will improve the MHSR method by studying the influence of anomaly size on it.

Q4: Ln 35: The value of frequency is missing.

A4: In Line 35, the intension of “ultrasound below kHz” is “ultrasound below the kHz level”. Thank you for your comments, I am aware that the previous statement was problematic, so I have revised that.

Q5:  Ln 117: “In an ideal scenario” this is vague. Please provide explanation/context. Are the authors referring to no attenuation? Also, since this work is related to pipe, the authors might want to consider talking about the presence of fluid(pressure) internally, and how does it affect their approach

A4: In Line 117, the ideal scenario means that the excitation/receiving transmission coefficient, electromagnetic-mechanical coefficient, and mechanical-electromagnetic coefficient in ultrasonic guided wave inspection are all 1. In simple terms, the ideal situation is that the energy transfer process of ultrasonic guided waves is lossless.

Regarding whether the internal fluid of pipes or pressure conditions within pipes would affect the MHSR method proposed in the paper, I would like to clarify that I have conducted related study in a paper named “Sparse reconstruction of ultrasonic guided wave signals of fluid-flled pipes by multistrategy hybrid DBO-OMP using dispersive Hanning-windowed chirplet model”published in "Measurement" this year, which corresponds to Reference [37] in the manuscript. The screenshot of the first page of the paper is as follows:

In the reference [37], The influence of the internal fluid in the pipe on the dispersion curve of ultrasonic guided waves was analyzed, and sparse reconstruction of ultrasonic guided wave signals in a fluid-filled pipe was carried out, of which the related figures are shown as follows. The conclusion is that the internal fluid in the pipe will cause mode coupling in the dispersion curve, and the dispersion parameters have changed compared with those of vacant pipe. Therefore, in order to ensure the accuracy of the reconstruction, sparse reconstruction cannot be carried out only based on the theoretical dispersion curve of the empty pipe, but should be based on the actual pipe's dispersion curve.

In this manuscript, MHSR uses the ST-SWA method to obtain more realistic pipe dispersion curves, which provides a more precise dispersion parameter search space for subsequent sparse reconstruction, thereby improving the accuracy of sparse reconstruction. Therefore, I believe that MHSR has the ability to accurately sparse reconstruction of ultrasonic guided wave signals for fuid-filled pipes or pressure pipes.

Of course, this manuscript did not conduct ultrasonic guided wave testing on fuid-filled pipes and pressure pipes. I will set up the relevant experimental conditions for this research in the future.

Q6:   Ln 127: “The level of agreement…” I do not see a reference to a figure. Is this talking about Fig2a?

A6: Thank you for your comments. “The level of agreement…” is talking about Figure. 2a, and I have added the reference.

Q7:  Ln 182: “NDE” or “NDM”?

A7: Thank you for your comments. “NDE” is the incorrect description, I apologize for the basic typographical error. I have changed NDE throughout the manuscript to NDM.

Q8:  Ln 224-226: Please correct these lines.

A8: Thank you for your comments. I have revised Line 224-226 from “As the measurement position advanced, the propagation distance of the UGW increased, causing the energy of the dispersive L(0,1) mode with weak dispersion to gradually diffuse in the time domain, resulting in broadening of the wave packet.” to “As the measurement position advanced, the propagation distance of the UGW increased, leading to a gradual diffusion of energy in the time domain for the dispersive L(0,1) mode with weak dispersion, resulting in temporal broadening of the wave packet.” I sincerely hope the above changes will meet your satisfaction.

Q9:    Ln 230,232: Hard to follow the comparison made for fig.5 using horizontal and vertical.

A9: Based on your suggestion, I realized that adding “Horizontally and Vertically” in the text was unnecessary and might actually make it harder to follow, so I deleted “Horizontally and Vertically”.

Q10:  fig.5: Has a lot of information that is not labeled. For example, fig5(a-d), are these the k-f curves for all 3 SNR. If yes, please annotate appropriately. Also, there are only two color bars for three rows of plots. (see comment 1)

A10: Based on your suggestion, I have added SNR annotation and color bars for Figure. 5, and the revised Figure is shown as follows:

Q11:   Ln 254: “NDE” or “NDM”?

A11: Thank you for your comments. “NDE” is the incorrect description, I apologize for the basic typographical error. I have changed NDE throughout the manuscript to NDM.

Q12: Fig6: Please see comment 1

A12: Thank you for your comments. I have enlarged Figure 6 to enhance the clarity of the details depicted in the image.

Q13:   Table 1 & Table 2: Could you please be more specific on how the error is calculated? Axial error seems intuitive but how do the authors define circumferential error? For example, in fig.9a, I see two bands where the max is uniform along d­ϑ for each of the D1 and D2

A13: Thank you for your comments. The location of the defect is obtained by taking the maximum pixel value of all nodes in the imaging area. Since the nodes have axial and circumferential position information, the node with the maximum pixel value is identified as the defect. The errors in axial and circumferential directions between the actual defect location and the calculated node location can be used to obtain Table 1 and Table 2.

In Figure 9a, due to the limited number of sensors and the fact that the circumferential distance is much smaller than the axial distance, the DAS image has lower resolution in the circumferential direction, and usually requires thresholding to distinguish the lateral position of defects.

Following your recommendation, I have incorporated a description of the process for calculating defect localization error in Line 404-410 as follows: “Identify the node with the highest pixel value as the defect, extract the axial and circumferential coordinates of this node, and calculate the estimated defect positioning errors of above methods by subtracting the actual coordinates, of which the results are presented in Table. 1. It is important to note that, due to the presence of two defects in the imaging results depicted in Figure 11, the imaging area should be partitioned into two segments for separate computation of defect location errors.

The corrections in the paper and the responds to the comments are described above, we have tried our best to improve the manuscript and made some changes in the manuscript, and hope the correction will meet with approval. Once again, thank you for your comments and suggestions.

Reviewer 2 Report

Comments and Suggestions for Authors

1.   The source of Formula (1) should be indicated.

2.   Why the perturbated coefficient δ is set to 0.1, 0.05, -0.05, and -0.1 respectively ?  Provide a reasonable explanation.

3.   The cg in lines 146-148 should be modified to cg.

4.   Validation of Experimental UGW Signals, Only the estimated defect position errors of experimental signals by DAS EVD RAPID MUSIC MOMP methods are provided, the essential reason for the high accuracy of MHSR method has not been theoretically analyzed.

Comments on the Quality of English Language

Minor editing of English language required

Author Response

The complete version of the reply can be downloaded as an attachment.

Dear Editors and Reviewers:

Thank you for your letter and the comments concerning our manuscript entitled “A multi-strategy hybrid sparse reconstruction method based on spatial-temporal sparse wave number analysis for enhancing pipe ultrasonic guided wave anomaly imaging” (sensors-3117990). Those comments are all valuable and very helpful for revising and improving our paper, as well as the important guiding significance to our researches. We have studied comments carefully and have made correction which we hope meet with approval. Revised portion are marked in yellow in the paper. The corrections in the paper and the responds to the comments are as follows:

Q1: The source of Formula (1) should be indicated.

A1: Thank you for your comments. According to your suggestion, I have marked the source of Formula (1), and its reference number is 37: [37] Tang B., Wang Y., Gong R., Zhou F. Sparse reconstruction of ultrasonic guided wave signals of fluid-filled pipes by mul-tistrategy hybrid DBO-OMP using dispersive Hanning-windowed chirplet model. Measurement, 2024, 231: 114648.

This paper is my previous research. The derivation of Formula (1) in Reference [37] is shown as follows:

Figure 1. Screenshot of Reference [37]

The model name in Reference [37] is “Third-order dispersive approximation based Hanning- windowed chirplet (TDAHWC)”, It is only for the case when the excitation signal is Hanning-windowed chirplet. In the draft, Equation (1) does not clarify the signal type of the excitation signal. It only refers to the excitation spectrum as Ye(ω) in general without expanding it in the form of an analytical expression. Therefore, Equation (1) can be widely applied to different excitation signal types. The nonlinear dispersion model corresponding to draft Equation (1) is essentially the same as the TADHWC model in Reference [37], both using the third-order Taylor expansion of k(ω) instead of k(ω).

Take Formula 1 in the manuscript as an example to illustrate the specific operation process. First, on the basis of the spectrum  of the ultrasonic guided wave signal in the ideal case, replace k(ω) with , and obtain:

(1)

Perform the Inverse Fourier Transform (IFFT) on the above equation to obtain the time-domain ultrasonic guided wave signal as follows:

(2)

In order to clarify that Formula 1 in the manuscript is the time-domain expression of the nonlinear dispersion model, I added "in the time domain" in the last sentence of the first paragraph of Section 3: “which is named the nonlinear dispersive model (NDM) in this paper and its expression in the time domain is as follows [37]:

Finally, I sincerely hope that my explanations can fulfill your expectations. Should you have any further inquiries in other respects, please do not hesitate to notify me and I will exert my utmost efforts to respond and make necessary amendments.

Q2: Why the perturbated coefficient δ is set to 0.1, 0.05, -0.05, and -0.1 respectively ?  Provide a reasonable explanation.

A2: Thank you for your comments. Regarding the setting of the perturbation coefficient δ, I referred to the study of Kim, which is Reference [27]: [27] Kim H., Yuan F. G. Adaptive signal decomposition and dispersion removal based on the matching pursuit algorithm using dispersion-based dictionary for enhancing damage imaging. Ultrasonics, 2020, 103: 106087. In Reference 27, the description of the range of the perturbation coefficient is shown in the following figure.

Figure 2. Screenshot of Reference [27]

In Reference [27], the range of the perturbation coefficient δ is [-5%, 5%], that is, [-0.05, 0.05], and the dispersion perturbation is achieved by . Thank you for your questions about the perturbation coefficient, which also enabled me to promptly discover that the expression for the dispersion parameter set in the draft was incorrect. In the previous version of the manuscript, the expression for the dispersion perturbation parameter was , which was a writing error. I have corrected it in the revised version, and the correct expression should be . Compared with the range of the perturbation coefficient δ in Reference [27], the range of the perturbation coefficient δ in the manuscript is larger, which is [-0.1, 0.1], with the aim of examining whether MHSR has the ability to accurately reconstruct the ultrasonic guided wave signal in the case of severe dispersion perturbation.

Q3: The cg in lines 146-148 should be modified to cg.

A3: Thank you for your comments. I'm very sorry that I didn't conduct a careful proofreading after writing the draft, resulting in such a low-level writing error in the draft. I have changed "cg" to "cg" in lines 146 to 148 and carefully checked the entire manuscript to ensure that no similar writing errors occur in the manuscript.

Q4: Validation of Experimental UGW Signals, Only the estimated defect position errors of experimental signals by DAS EVD RAPID MUSIC MOMP methods are provided, the essential reason for the high accuracy of MHSR method has not been theoretically analyzed.

A4: Thank you for your comments. Based on your question, I came to the realization that the analysis of MHSR achieving high accuracy in Part 6 was incomplete. Consequently, I have made amendments to it. The specific contents are presented as follows:

“The VMD defect echo signals were processed by DAS, EVD, RAPID, MUSIC, MOMPI, and MHSRI, and the anomaly imaging results are depicted in Figure. 14 with triangles representing the actual defect positions. Subsequently, the estimated defect positions obtained from these imaging methods were extracted and the errors between the estimated and actual positions were calculated in Table. 3. In comparison to the synthesized noisy disturbed signal, it was observed that due to imperfect experimental conditions, the experimental signal contained higher noise content which led to reduced resolution of defects in DAS, EVD, and MUSIC imaging. Furthermore, based on k(f) extracted by ST-SWA analysis revealed relatively close resemblance between actual and theoretical frequency dispersion characteristics of the pipeline, indicating that conventional imaging methods yielded relatively accurate estimations of defects compared with those of dispersion disturbance in Section 5.3. However, the enhancement effect is constrained, and the defect positions obtained exhibit axial and circumferential error ranges of [21 mm, 115 mm] and [51 mm, 111 mm], respectively.

Due to the sparse reconstruction capability of MOMP and MHSR for multi-mode dispersion measured signals, along with the ability of dispersion compensation to convert the reconstructed signal into a single-mode non-dispersive signal, coherent noise effects can be completely eliminated. Consequently, MOMPI and MHSRI exhibit high defect resolution. However, as discussed in Section 5.3, due to its reliance on the theoretical frequency k(f) for reconstruction, MOMP exhibits relatively large reconstruction errors for propagation distance d. As a result, the positioning accuracy of MOMPI for defects is not optimal, with an average positioning error of (20 mm, 15 mm). In contrast to MOMP, MHSR utilizes ST-SWA to estimate the dispersion curve and obtain k(f) closer to actual values than theoretical solutions. This contributes to more accurate reconstruction results for MHSR and lower error in propagation distance d. Consequently, the positioning accuracy of MHSRI for defects is enhanced with an average positioning error of (11.5 mm, 11 mm). ”

The corrections in the paper and the responds to the comments are described above, we have tried our best to improve the manuscript and made some changes in the manuscript, and hope the correction will meet with approval. Once again, thank you for your comments and suggestions.

Reviewer 3 Report

Comments and Suggestions for Authors

This manuscript presents a methodology for damage imaging in steel pipes. The methodology appears to initially process the received signals before applying damage imaging algorithms to reconstruct images of the defects. The manuscript requires major improvements to be considered for publication. The main concerns and recommendations are as follows:

  1. The novelty of the work is not clear. What is the purpose of multi-strategy hybrid sparse reconstruction? Is it to remove noise and find a specific mode's dispersion curve and velocity? If yes, how does this method compare to existing damage imaging algorithms?
  2. As mentioned in the manuscript, accurate dispersion curves are important for some damage imaging algorithms. It is recommended to include the following experimental approaches for UGW dispersion curve reconstructions in the references to improve the literature review:
    • [1] L. Zeng et al., "Determination of Lamb wave phase velocity dispersion using time–frequency analysis," Smart Mater. Struct., vol. 28, no. 11, p. 115029, Nov. 2019 [Online]. Available: 10.1088/1361-665X/ab47e1.
    • [2] M. Barzegar et al., "Experimental Estimation of Lamb Wave Dispersion Curves for Adhesively Bonded Aluminum Plates, Using Two Adjacent Signals," IEEE Trans. Ultrason. Ferroelectr. Freq. Control, vol. 69, no. 6, pp. 2143–2151, Jun. 2022 [Online].
    • [3] Qi Chen et al., "High-resolution Lamb waves dispersion curves estimation and elastic property inversion," Ultrasonics 115 (2021): 106427.
  3. The MOMP method is not well explained and requires further explanations with examples of signals, its inputs, and outputs. The purpose of the method in this study needs to be clarified.
  4. Subsection 4.3. Sparse reconstruction imaging needs comprehensive explanations with examples. In its current form, it is not easy to follow. What do the superscripts e and r in line 214 represent? How does the dictionary D represent an image? What is  in Figure 3?
  5. Similarly, subsection 5.3. Pipe sparse reconstruction imaging requires elaboration. What function of noise is added to signals? What is the equation used?
  6. There are too many damage imaging algorithms mentioned, including DAS, EVD, RAPID, etc., but they need explanations. For example, the RAPID algorithm provides elliptical distributions between pairs of sensors. Which pairs of sensors produced the image in Figure 9(c)? Moreover, the RAPID method as stated uses comparisons to baseline data. How was this obtained? This applies to other methods as well. For example, what is the Eigenvalue Decomposition (EVD) method for imaging?
  7. The authors compared MOMP and MHSR to state-of-the-art damage imaging methods. However, the manuscript lacks explanations on the differences between these methods and requires further analysis of why each of these methods fails or succeeds.

Minor comments:

  • NDE should be written in full as "Nondestructive Evaluation."
  • Elastic modulus is better to be specified as 217 GPa.
  • Clarify what is meant by "high-resolution defect."
Comments on the Quality of English Language

The English Language Needs minor revision.

Author Response

The complete version of the reply can be downloaded as an attachment.

Dear Editors and Reviewers:

Thank you for your letter and the comments concerning our manuscript entitled “A multi-strategy hybrid sparse reconstruction method based on spatial-temporal sparse wave number analysis for enhancing pipe ultrasonic guided wave anomaly imaging” (sensors-3117990). Those comments are all valuable and very helpful for revising and improving our paper, as well as the important guiding significance to our researches. We have studied comments carefully and have made correction which we hope meet with approval. Revised portion are marked in yellow in the paper. The corrections in the paper and the responds to the comments are as follows:

Q1: The novelty of the work is not clear. What is the purpose of multi-strategy hybrid sparse reconstruction? Is it to remove noise and find a specific mode's dispersion curve and velocity? If yes, how does this method compare to existing damage imaging algorithms?

A1: Thank you for your comments. Based on your question, I will provide answers one by one. Here are my specific responses:

(1) "The novelty of the work is not clear." I apologize for not clearly stating the novelty of the work in the abstract and introduction of the manuscript. I have carefully reviewed it, and the innovative aspects include the following:

1) To address the problem that the conventional ultrasonic guided wave defect imaging method has low defect positioning accuracy due to its reliance on the dispersion curve theoretical solution, this paper proposes using spatial-temporal sparse wavenumber analysis (ST-SWA) to estimate the dispersion curve of the pipeline in actual ultrasonic guided wave detection. Through the analysis of the synthetic signal of ultrasonic guided wave in Section 5.1, the results show that ST-SWA can obtain high-resolution and accurate dispersion curves. The estimated dispersion curve that is similar to the actual dispersion curve can provide more accurate group velocity cg, which is beneficial to the accuracy of defect positioning in defect imaging, no matter which ultrasonic guided wave defect imaging method is used.

2) During ultrasonic guided wave inspection, coherent noise may appear in the measured signal due to imperfect detection conditions, mode conversion, and the intrinsic characteristics of ultrasonic guided waves (dispersion and multimodality). This is not conducive to obtaining a high signal-to-noise ratio and accurate defect location in ultrasonic guided wave imaging. Therefore, many scholars use sparse reconstruction methods to reconstruct ultrasonic guided wave signals to eliminate the influence of coherent noise. Sparse reconstruction methods essentially involve linear combinations of atoms in a pre-constructed redundant dictionary to achieve the representation of the input signal y, i.e., y = Dx + e, where D is the dictionary, x is the reconstruction coefficient, e is the error, and  is the reconstructed signal. Therefore, the basis functions used to construct the dictionary and the method for solving the reconstruction coefficient are the key factors that determine the reconstruction accuracy of ultrasonic guided wave signals.

First, regarding the basis functions of the dictionary, i.e. the ultrasonic guided wave signal model, the mainstream model currently used is Gabor and Chirplet, but they cannot represent the asymmetric envelope and nonlinear instantaneous frequency characteristics of ultrasonic guided wave signals, which is not conducive to the accurate reconstruction of ultrasonic guided wave signals. Second, regarding the method for solving the reconstruction coefficient, the commonly used methods include the matching pursuit, orthogonal matching pursuit, basis pursuit, and compressive sampling matching pursuit, etc. In order to improve the reconstruction accuracy, a smaller model parameter step size and a larger number of dictionary atoms are often needed, which will lead to an increase in the computational cost.

Therefore, this paper proposes using the Modified-OMP (MOMP) algorithm based on the nonlinear dispersion model (NDM) for sparse reconstruction of multi-mode dispersive ultrasonic guided wave signals. It is worth noting that the NDE and MOMP methods were proposed in the author's previous research and published in the "Measurement" journal under the ELSEVIER umbrella. This paper also cites them, as shown in Reference [37] on the first page of the paper, with a screenshot of the reference list as follows:

Figure 1 Screenshot of Reference [37]

In Reference [37], NDM was explained in detail as the Third-order dispersive approximation based Hanning-windowed chirplet model (TDAHWC). In simple terms, the principle of NDM is to replace the wave number k(ω) with its third-order Taylor expansion, thereby achieving controllable dispersion parameters while ensuring the non-symmetric envelope and nonlinear instantaneous frequency characteristics of the ultrasonic guided wave signal. Figure 2 is a screenshot of the reconstruction effect of the TDAHWC model from Reference [37], which compares the reconstruction effects of the conventional models (Gabor, Chirplet, HWC), improved models (AGWC, NHWC), and the proposed TDAHWC model on the ultrasonic guided wave signal from both the time domain signal envelope and the time-frequency domain instantaneous frequency (IF) perspectives, concluding that the TDAHWC model is feasible for representing the ultrasonic guided wave signal. In order to avoid the possibility of duplicate research on the same model leading to an increase in the paper's duplication rate, the manuscript only provides a brief explanation of it and does not provide a detailed discussion. Additionally, the model name in Reference [37] is quite long and difficult to write and read, so the manuscript abbreviates it to NDM.

Figure 2 Screenshot of the TDAHWC reconstruction effect as described in Reference [37].

In Reference [37], MOMP was expanded upon and explained as dung beetle optimization based orthogonal matching pursuit (DBO-OMP). Since MOMP is a previous research result (Reference [37]), this paper only applies it and, to avoid a high duplication rate in the paper, I did not provide a detailed explanation of MOMP in the manuscript. However, you mentioned in Question 3 that MOMP was not fully explained, and I also realized that this part of the explanation is very necessary, so I added a description of MOMP in the revised answer. See Answer 3. Figure 3 is a screenshot of the reconstruction effect of DBO-OMP from Reference [37].

Figure 3 Screenshot of the reconstruction results obtained from DBO-OMP as described in Reference [37].

3) In ultrasonic guided wave imaging, considering the importance of dispersion curve to the accuracy of defect location and the unfavorable influence of coherent noise on imaging resolution, this paper proposes using the ST-SWA-based multi-strategy hybrid sparse reconstruction (MHSR) to process the array measurement signals, thereby achieving high-resolution and high-precision ultrasonic guided wave sparse imaging. The algorithm flowchart of MHSR is shown in Figure 3 of the manuscript.

Figure 4. Figure 3 in the manscript (algorithm flow of MHSR)

The working principle of MHSR mainly includes three steps. The first step is to use the ST-SWA to obtain an estimated dispersion curve of the actual ultrasonic guided wave, compared with the theoretical solution, the estimated solution is closer to the actual solution; the second step is to perform a third-order Taylor expansion on the estimated dispersion curve to obtain dispersion parameters, based on the estimated dispersion parameters, MOMP can achieve sparse reconstruction of multi-mode dispersive ultrasonic guided wave signals with lower reconstruction accuracy; the third step is that since the reconstructed signal is the superposition of multi-mode dispersive ultrasonic guided wave signals, the output of MOMP includes not only the reconstructed signal but also the dictionary used for reconstruction and the reconstruction coefficient x. Using the output results, the non-dispersion and dispersion compensation are performed sequentially, and the high signal-to-noise ratio of the single-mode non-dispersion reconstructed signal obtained finally is very high, which is beneficial to improving the resolution and positioning accuracy of defects in ultrasonic guided wave imaging.

(2) Regarding the question "Is it to remove noise and find a specific mode's dispersion curve and velocity?", my answer is yes. MHSR does utilize ST-SWA to obtain an estimated dispersion curve that is more in line with actual conditions than the theoretical dispersion curve, and it eliminates the effects of coherent noise through operations such as MOMP reconstruction, deconvolution, and dispersion compensation.

(3) In response to the question "If yes, how does this method compare to existing damage imaging algorithms?", I would like to clarify that MHSR serves as a preprocessing method for sparse reconstruction imaging. The effectiveness of sparse reconstruction imaging is closely linked to the MHSR signal. Upon reviewing your feedback, I recognize that the wording in the manuscript was inappropriate as it conflated the concepts of MHSR and sparse reconstruction imaging based on MHSR. To distinguish between them, I will simply refer to sparse reconstruction imaging based on MHSR as MHSRI and make necessary revisions in the manuscript accordingly. Based on these modifications, MHSRI can be compared with conventional ultrasonic guided wave imaging methods. Regardless of their underlying principles, conventional ultrasonic guided wave imaging methods utilize multi-mode dispersive ultrasonic guided wave signals as array signals; whereas, MHSRI employs single-mode non-dispersive ultrasonic guided wave signals which ensures superior imaging resolution. Furthermore, the dispersion parameters used in MHSRI are derived from ST-SWA estimation, providing greater accuracy in defect location compared to theoretical dispersion parameters used in conventional ultrasonic guided wave imaging.

Q2: As mentioned in the manuscript, accurate dispersion curves are important for some damage imaging algorithms. It is recommended to include the following experimental approaches for UGW dispersion curve reconstructions in the references to improve the literature review

  • [1] L. Zeng et al., "Determination of Lamb wave phase velocity dispersion using time–frequency analysis," Smart Mater. Struct., vol. 28, no. 11, p. 115029, Nov. 2019 [Online]. Available: 10.1088/1361-665X/ab47e1.
  • [2] M. Barzegar et al., "Experimental Estimation of Lamb Wave Dispersion Curves for Adhesively Bonded Aluminum Plates, Using Two Adjacent Signals," IEEE Trans. Ultrason. Ferroelectr. Freq. Control, vol. 69, no. 6, pp. 2143–2151, Jun. 2022 [Online].
  • [3] Qi Chen et al., "High-resolution Lamb waves dispersion curves estimation and elastic property inversion," Ultrasonics 115 (2021): 106427.

A2: Thank you for your comments. Following your guidance, I believe that these three papers can enhance the introductory content, and I have incorporated pertinent information into the revised introduction:“Barzegar et. al [33] applied the classical 2D-FFT to get wave-number distribution of Lamb UGWs, and the comparison of estimated and theoretical dispersion curves showed the low error. Zeng et. al [34] recommended the adoption of short-time chirp-Fourier trans-form, ridge tracking and Vold–Kalman filter to obtain the Lamb wave dispersion curves. Chen et. al [35] extracted the dispersion curves via rotation invariant technique (ESPRIT), which is inducive to the model-based elastic property estimation.” (Page 3, P97-103)

Q3: The MOMP method is not well explained and requires further explanations with examples of signals, its inputs, and outputs. The purpose of the method in this study needs to be clarified.

A3: Thank you for your comments. I did not provide a detailed explanation of MOMP in the previous manuscript because I considered it to be part of my prior research, with this paper merely utilizing it. However, your insight has prompted me to recognize that without a clear elucidation of MOMP, the fundamental principles of MHSR may not be comprehensible. Consequently, I have made substantial revisions to Section 4.2, specifically incorporating the following content:

4.2. Multi-strategy hybrid sparse reconstruction

Based on the k(f) from ST-SWA, the reconstruction of multi-mode dispersive UGW signals can be achieved by the modified OMP (MOMP) method proposed in the previously published paper [37]. Given the large size dictionary is essential for the high reconstruction accuracy of the sparse reconstruction, MOMP employ the dung beetle optimization (DBO) algorithm [42] instead of the greedy search process in OMP, which can rapidly and effectively identify the globally optimal NDM parameters. The dung beetle position Para represents the NDM model parameters, , , where i denotes the i-th mode and K denotes the number of modes for reconstruction. The upper and lower bounds of the dung beetle position can then be expressed as  and , where,. The fitness function can be expressed as

(3)

(4)

where y is the input signal, ,  is the dispersive atom dereived by substituting  into NDM (Formula 1). DBO has be designed by ball-rolling, dancing, breeding, foraging and stealing processes, they correspond to different dung beetle position update strategies, which are explained in detail in Xue's research [42]. The pseudo-code for MOMP is as follows.

The inputs are the population size N, the maximum iteration number I, the sparsity τ, and the bounds Ub and Lb. The outputs are the reconstruction signal , the global optimal NDM parameter set , the dispersive dictionary , and the reconstruction coefficient . In addition, the Bounds function is to control the position not to exceed the bounds Ub and Lb.

Algorithm 2 MOMP

Input: N, I, τ, Ub, Lb

Initialize: , ,

Output: , , ,

for k ←1 to τ do

Initialize: 

while (iI) do

for j ←1 to N do

if j == ball-rolling dung beetle then δ=rand(1);

if δ <0.9 then Update  by ball-rolling way;

else Update  by dancing way;

end if

;

end if

if j == brood ball then

Update  by breeding way, ;

end if

if j == small dung beetle then

Update  by foraging way, ;

end if

if j == thief then

Update  by stealing way, ;

end if

end for

if  then Update Para(i);

end if

i=i+1;

end while

,, , , ,

end for

return ,, ,

It is worth noting that the initial boundaries of MOMP are set based on the theoretical dispersion parameters. Thanks to the estimated dispersion curves by ST-SWA, the search space defined by the upper and lower bounds can be more precisely specified, thereby contributing to enhanced accuracy in reconstruction. To mitigate the impact of coherent noise, the reconstructed signal should be process by the dispersion compensation, of which the multi-mode dispersive signal can be convert to signal-mode non-dispersive signal. Considering the output of MOMP also includes the NDM parameter set , of which the values of k1 of different mode can be extract to realize the dispersion compensation. The dispersion compensation can be achieved by the following operation:

(5)

(6)

Where n and m represents the maximum circumferential order and family number, and the subscript (i,j) corresponds to the order and family number of the given mode. The method that includes ST-SWA, MOMP, and dispersion compensation is called multi-strategy hybrid sparse reconstruction (MHSR), and its flowchart is shown in Figure. 3.

Figure 3. Flowchart of multi-strategy hybrid sparse reconstruction.”

Therefore, based on the aforementioned revised Section 4.2, the purpose of MOMP is: “Based on the k(f) from ST-SWA, the reconstruction of multi-mode dispersive UGW signals can be achieved by the modified OMP (MOMP) method proposed in the previously published paper [37].”The input and output of MOMP are:“The inputs are the population size N, the maximum iteration number I, the sparsity τ, and the bounds Ub and Lb. The outputs are the reconstruction signal , the global optimal NDM parameter set , the dispersive dictionary , and the reconstruction coefficient . In addition, the Bounds function is to control the position not to exceed the bounds Ub and Lb.”

Finally, I am grateful for your valuable suggestions, which have prompted me to recognize the significance of augmenting the theoretical framework of the MOMP algorithm. I trust that the aforementioned adjustments and response will align with your expectations.

Q4: Subsection 4.3. Sparse reconstruction imaging needs comprehensive explanations with examples. In its current form, it is not easy to follow. What do the superscripts e and r in line 214 represent? How does the dictionary D represent an image? What is  in Figure 3?

A4: Thank you for your comments. In response to your valuable feedback, I have made the following revisions and provided a detailed response:

(1) Regarding "Subsection 4.3. Sparse reconstruction imaging needs comprehensive explanations with examples." I have made significant changes to Section 4.3, specifically as follows:

4.3. Sparse reconstruction imaging

In addition to the conventional UGW anomaly imaging methods mentioned in the introduction, sparse reconstruction imaging, inspired by sparse reconstruction method, represents the effective enhancement in resolution. Xu et al. [43] proposed an anomaly imaging method based on weighted sparse reconstruction, which obtains pixel values at each position in the imaging area by solving the weighted sparse reconstruction problem. For a better understanding of the principles behind sparse reconstruction imaging, its process was illustrated in Figure. 4.

Figure 4. Flowchart of multi-strategy hybrid sparse reconstruction imaging.

Suppose that the number of excitation-receiving pairs used for UGW imaging is L. The hollow cylindrical pipe is unfolded into a planar rectangle, and the imaging area is set and discretized into N nodes. Since each node may be a reflection source, a dictionary D can be constructed that contains L×N signals, and the reconstruction coefficient x with a specified sparsity τ can be obtained by solving following problem:

(5)

Where σ is the standard deviation of the error term e (y=Dx+e), y is the matrix that contains L different measured signals, represented as y=[y1,y2,…,yL]T, and D is constructed in the following way:

(6)

Due to its capability in eliminating coherent noise and reconstructing single-mode non-dispersive signals, the measured signal y can undergo preprocessing using MHSR. Therefore, the atom gi,j in D can be constructed through Equation 6, where k1 of the desired mode can be extracted from  obtained by MHSR, and the propagation distance d can be calculated by , where dx and dy denote the two-dimensional coordinates of the node or sensor, the superscript e and r represent the excitation and receiving sensors, the subscript i and j represent the serial numbers of signals and nodes.

It is worth noting that the reconstruction coefficient x is an N×1 dimensional matrix, it should be rearranged according to the node distribution, and the absolute value of the rearranged matrix denotes the pixels in the two-dimensional imaging area. Due to the strong sparsity of x, with only a small number of non-zero pixels, it is feasible to transform the two-dimensional imaging results into a three-dimensional display for clear visualization of defects.”

The idea behind the above modifications is as follows:

1) First, by conducting research on sparse overlapping image reconstruction, we introduce sparse overlapping image reconstruction as the imaging method in this paper;

2) Second, we draw a flowchart of the principle of sparse overlapping image reconstruction to facilitate reading and understanding;

3) Finally, we explain the principle of sparse overlapping image reconstruction through theoretical formulas and explain how to convert the sparse reconstruction coefficient x from an N×1 dimensional matrix to a two-dimensional and three-dimensional image.

(2) In response to What do the superscripts e and r in line 214 represent? How does the dictionary D represent an image? “What is A in Figure 3?” made me realize that the manuscript was lacking in its description of symbols and variables. Based on your feedback, I have made modifications to the manuscript. The superscript e and r represent the excitation and receiving sensors, respectively. Dictionary D is used to obtain the reconstruction coefficients x, and the reconstruction coefficients x are used to represent the image. The A in the previous manuscript corresponds to the modified version of x, which corresponds to the reconstruction coefficients x, and I have changed it in Figure 3. Additionally, I have modified Figure 3 to make it easier to read and understand.

Thank you for your valuable feedback, and I sincerely apologize for the errors identified in Section 4.3 of the manuscript. I am committed to ensuring that my revisions align with your expectations. Should you believe that further refinement is necessary, please do not hesitate to inform me at your earliest convenience, and I will diligently incorporate your suggestions.

Q5: Similarly, subsection 5.3. Pipe sparse reconstruction imaging requires elaboration. What function of noise is added to signals? What is the equation used?

A5: Thank you for your comments. In response to "Subsection 5.3. Pipe sparse reconstruction imaging requires elaboration," I have also made extensive modifications to Section 5.3, particularly in the analysis of the differences between conventional ultrasonic guided wave imaging methods and the MOMPI and MHSRI imaging methods proposed in this paper. Specifically:

“To distinguish, the sparse reconstruction images based on MHSR and MOMP are denoted as MHSRI and MOMPI, respectively. The anomaly imaging results of DAS, Eigenvalue decomposition (EVD), RAPID, MUSIC, MOMPI, and MHSRI are presented in Figure. 11; where triangles denote actual defect positions. The computed defect positioning errors for each method are presented in Table. 1. Due to the perturbation coefficient δ of 0.1 in the synthesized measurement signals, DAS, EVD, RAPID, and MUSIC based on the theoretical dispersion curve exhibits diminished accuracy in defect location. Especially, the axial defect errors of above methods are very large, varying from 111.5 mm to 180.5 mm. The circumferential defect error is only 1 mm for MUSIC, which is deemed satisfactory, whereas those of other methods range from 23 mm to 34 mm. The limitation in achieving high defect location accuracy using the above methods stems from the substantial disparity between the theoretical dispersion curve and its actual value, as well as the constrained number of sensors which hinders enhancements in location accuracy. In addition, as the measured signal contains both coherent and incoherent noise, the imaging method described above exhibits limited defect resolution. Therefore, threshold processing is essential to achieve satisfactory imaging results.

Compared to conventional UGW imaging methods, both MOMP and MHSR perform a search and optimization of dispersion parameters, resulting in dispersion parameter values that closely approximate the actual values. Consequently, the positioning accuracies of MOMPI and MHSRI are significantly enhanced. In particular, MHSR, which has pre-calculated dispersion curves using ST-SWA, possesses a more constrained and precise dispersion parameter search space compared to MOMP. Therefore, MHSRI shows a reduced defect location error of only (3 mm, 6.5 mm). Furthermore, as the input to MHSRI consists of the single-mode non-dispersive signals with dispersion compensation, both coherent and non-coherent noise are effectively eliminated. Consequently, only a limited number of pixel values exhibit non-zero in MHSRI imaging, indicating exceptionally high defect resolution.”

Additionally, regarding the noise added to the synthetic signals, I would like to clarify that the noise in this paper mainly consists of coherent noise and non-coherent noise. Coherent noise is non-target ultrasonic guided wave, and non-coherent noise is the interference signal caused by the detection environment, which is represented by Gaussian noise in this paper. Regarding the addition of non-coherent noise, I mentioned it in line 303, "By introducing random Gaussian noise as non-coherent noise, noisy perturbed signals (NPS1-NPS4) were generated." Gaussian noise is easy to add in MATLAB and is a commonly used form of non-coherent noise, so I don't think it's necessary to explain its formula in the manuscript.

Q6: There are too many damage imaging algorithms mentioned, including DAS, EVD, RAPID, etc., but they need explanations. For example, the RAPID algorithm provides elliptical distributions between pairs of sensors. Which pairs of sensors produced the image in Figure 9(c)? Moreover, the RAPID method as stated uses comparisons to baseline data. How was this obtained? This applies to other methods as well. For example, what is the Eigenvalue Decomposition (EVD) method for imaging?

A6: Thank you for your comments. In accordance with your guidance, I present the following response:

(1) In response to There are too many damage imaging algorithms mentioned, including DAS, EVD, RAPID, etc., but they need explanations. In this paper, these ultrasonic guided wave imaging methods are used to highlight the advantages of MHSRI in terms of imaging resolution and defect location accuracy. Among these methods, DAS and RAPID are both classic imaging methods, and their related theories are very classic and complete. I also considered whether to explain their theories in detail in the manuscript, but I think that it is more important to explain the MHSRI method clearly. Of course, I studied all the imaging methods mentioned in the manuscript work, and their theories are explained in detail in the corresponding reference literature.

1) Reference [27] provides a theoretical description of DAS as shown in the following figure.

2) Reference [29] provides a theoretical description of RAPID as shown in the following figure.

3) Reference [31] provides a theoretical description of MUSIC as shown in the following figure:

4) Regarding EVD, I primarily referred to Wang Peng's doctoral dissertation from Harbin Institute of Technology: Ultrasonic Lead-Wave Testing Method for Fatigue Crack of Orthotropic Steel Bridge Deck with EVD Theory, where the relevant theory of EVD is shown in the following figure.

Clearly, the inclusion of detailed explanations for all the aforementioned methods would significantly increase the length of the manuscript. In consideration of this, while manuscripting, I focused on MHSRI rather than these imaging methods and thus refrained from providing detailed explanations.

(2) In response to For example, the RAPID algorithm provides elliptical distributions between pairs of sensors. Which pairs of sensors produced the image in Figure 9(c)? Moreover, the RAPID method as stated uses comparisons to baseline data. How was this obtained?” What I want to point out is that the ultrasonic guided wave mode used in this paper for detection is L(0,2), and the excitation and reception sensors are placed on the same side. Typically, the sensor arrangement in RAPID is as shown in the following figure:

The sensor arrangement depicted in the above figure results in a shift from the ultrasonic guided wave mode used for detection being L(0,2) to a mode more akin to Lamb waves in a flat plate. The sensor configuration outlined in this study is as follows: E1 serves as the excitation sensor encircling the pipe circumferentially, requiring only one sensor for achieving axially symmetric L(0,2) excitation. Eight receiving sensors are evenly distributed along the pipe circumference. While differing from the arrangement shown in the previous figure, the underlying principle of RAPID imaging remains unchanged.

RAPID necessitates a correlation analysis between the base signal and the damage signal, with no requirement to specify the source of the damage signal. In Section 5.3, the base signal is a synthetic waveform propagated over a distance of 2 m, whereas in Section 6, it is acquired through ultrasonic guided wave testing of pristine pipes.

(3) In response to “This applies to other methods as well. For example, what is the Eigenvalue Decomposition (EVD) method for imaging?”. Based on the above analysis, EVD imaging is actually based on DAS, where the input signal is processed with EVD to achieve noise reduction.

Finally, I would like to clarify that my intention is not to suggest that the above ultrasonic guided wave imaging methods are entirely ineffective, as many current studies on ultrasonic guided wave imaging utilize these methods. My emphasis lies in highlighting that when employing the aforementioned ultrasonic guided wave imaging methods, the group velocities utilized are based on theoretical ultrasonic guided wave curves. However, it is important to note that actual group velocities may deviate from theoretical values, leading to inaccuracies in defect positioning within the imaging results. Furthermore, due to coherent and incoherent noise present in ultrasonic guided wave signals, this imaging method does not undergo preprocessing of input data, resulting in noise interference and reduced defect resolution.

While I understand that this response may not fully meet your expectations, I maintain my belief that the primary focus of this article is centered around the proposed MHSRI. In consideration of your other feedback received, I am confident that the revised manuscript now provides a clearer elucidation of MHSRI. For readers interested in exploring existing ultrasonic guided wave imaging methods further, relevant reference literature has been provided in the introduction for independent access

Q7: The authors compared MOMP and MHSR to state-of-the-art damage imaging methods. However, the manuscript lacks explanations on the differences between these methods and requires further analysis of why each of these methods fails or succeeds.

A7: Thank you for your comments. Based on your suggestion, I first conducted a more detailed analysis of Section 5.2. The modifications made to Section 5.2 mainly involve MOMP and MHSR in terms of reconstruction of signals, reconstruction error of signals, dispersion compensation signals, and error in distance reconstruction. These modifications illustrate the advantage of MHSR over MOMP in accurately reconstructing multi-mode dispersive ultrasonic guided wave signals. The modified Section 5.2 is as follows:

5.2. Sparse reconstruction of dispersive perturbated signals

Considering potential deviations between the actual k(f) and its theoretical value, it is imperative to conduct dispersion perturbation analysis to validate the efficacy of the MHSR. NDM (Formula. 1) was employed to synthesize multi-mode dispersive UGW signals encompassing L(n,m) (n=0-2, m=1-2), with an energy ratio of 0.88:0.12, and the propagation distances of d1=1 m and d2=2 m. Assume the dispersion perturbation parameter set is , where  and perturbated coefficient δ is set to 0.1, 0.05, -0.05, and -0.1 respectively. By introducing random Gaussian noise as non-coherent noise, noisy perturbated signals (NPS1-NPS4) were generated, as illustrated in Figure. 7(a-d). Utilizing ST-SWA, the k(f) of NPS1-NPS4 were estimated and are illustrated in Figure 7(e-h). The figure demonstrates that the resolution of different modes is correlated with their energy in the time domain. Due to the low signal amplitude of L(n,1) in the time domain, which is nearly overwhelmed by non-coherent noise, its peak ridge resolution is notably limited. The k(f) of L(0,2), exhibiting the highest energy in the time domain and superior resolution, was isolated for extraction and subsequent calculation of its perturbated coefficient. Assuming that other modes share a similar disturbance coefficient as L(0,2), we obtained estimated k(f) for all modes as depicted in Figure. 7(e-h). The results indicate a high consistency between the estimated k(f) and trends observed in the peak ridges within the k-f distribution.

Figure 7. Noisy perturbated signals in time domain: (a) NPS1, (b) NPS2, (c) NPS3, and (d) NPS4, k-f distribution and estimated k(f) by ST-SWA: (e) NPS1, (f) NPS2, (g) NPS3, and (h) NPS4.

Figure 8. Reconstructed signals and non-dispersive components by MHSR: (a) NPS1, (b) NPS2, (c) NPS3, and (d) NPS4, dispersion compensated signals by MHSR and MOMP: (e) NPS1, (f) NPS2, (g) NPS3, and (h) NPS4.

Because the boundaries of MHSR are based on the estimated k(f) of ST-SWA, δ in  and  can be adjusted to a reduced value (0.02). While the boundaries of MOMP are based on the theoretical k(f), δ should be adjusted to a large value (0.1). In addition to the aforementioned variations in parameter settings, both MHSR and MOMP exhibit identical parameter configurations: dmax and dmin of Ub and Lb are 3 m and 0 m, population size N is 100, the maximum iteration number I is 200, and the sparsity τ is 2. Utilizing MHSR and MOMP for processing NPS1-NPS4, Figure. 8(a-d) displays the reconstructed signals and residual signals. It can be observed that both MHSR and MOMP demonstrate the capability to achieve satisfactory reconstruction results in the presence of non-coherent noise. However, compared with MHSR, it can be notice that the reconstructed signals of MOMP exhibits larger reconstruction errors, of which the reconstruction error ranges of MHSR and MOMP are [0.15, 0.33] and [0.29, 0.52]. Calculate the residual signals after the sparse reconstruction, as shown in Figure 7(e-h). It can be found that the reconstruction accuracy of MHSR remains high even in the presence of dispersion disturbances and non-coherent noise. The overall amplitude of the residual signal is minimal, and the reconstruction performance of the d1=1 m wave packets of NPS2 is suboptimal, primarily due to the elevated levels of non-coherent noise. However, the overall amplitude of the residual signals in MOMP is relatively high, particularly for NPS1, where the amplitude of the residual signal for L(n,1) in MOMP reaches 0.1, which is attributed to the broader bounds derived from theoretical k(f).

Due to the MHSR reconstructed signals are multi-mode dispersive UGW signals, it is necessary to perform de-dispersion as a preliminary step, Figure. 9(a-d) presents the non-dispersive components of MHSR signals. Utilizing Formula. 5, the non-target modes are compensated into the target mode L(0,2) based on the multi-mode non-dispersive signals. To emphasize further on the reconstruction accuracies of MHSR and MOMP, the dispersion compensated signals of MHSR and MOMP are illustrated in Figure. 9(e-h), of which the theoretical dispersion compensated signals are also depicted. The discrepancy in reconstruction errors between MHSR and MOMP are exemplified with respect to the dispersion compensated signals. The MHSR compensated signals closely aligns with the theoretical signals, exhibiting only minor discrepancies in amplitude. In contrast, the MOMP compensated signals not only displays significant deviations in amplitude from the theoretical signals but also exhibits substantial disparities in arrival times.

Figure 9. Non-dispersive components of MHSR signals: (a) NPS1, (b) NPS2, (c) NPS3, and (d) NPS4, dispersion compensated signals by MHSR and MOMP: (e) NPS1, (f) NPS2, (g) NPS3, and (h) NPS4.

Table 1. Reconstructed propagation distance errors of MHSR and MOMP.

NPS1

NPS2

NPS3

NPS4

d1 (mm)

d2 (mm)

d1 (mm)

d2 (mm)

d1 (mm)

d2 (mm)

d1 (mm)

d2 (mm)

MHSR

4.24

11.79

4.26

7.05

0.96

1.44

5.15

8.63

MOMP

10.82

24.48

9.81

28.31

3.63

17.28

6.29

16.73

The reconstruction accuracies of the dispersion compensated signals are largely contingent upon the precision of the reconstructed dispersion parameters, with particular emphasis on the influence of the reconstructed propagation distance d. The reconstructed propagation distance errors of MHSR and MOMP are calculated, which are shown in Table. 1. In comparison to MOMP, the search space for the dispersion parameters in MHSR is more compact and precise, resulting in a significantly reduced reconstruction error for the propagation distance. Particularly, the errors for NPS3, d1 and d2 are merely 0.96 mm and 1.44 mm. However, the overall reconstruction error level for MOMP is relatively high, particularly with respect to NPS2, d1 and d2 reaching 9.81 mm and 28.31 mm.”

I hope the above changes meet your satisfaction. If you still think changes are needed, I will listen to your opinion and make further modifications.

Q8: NDE should be written in full as "Nondestructive Evaluation."

A8: Thank you for your comments, I have revised NDE to "Nondestructive Evaluation."

Q9: Elastic modulus is better to be specified as 217 GPa

A9: Thank you for your comments, I have written the modulus of elasticity as 217 GPa.

Q10: Clarify what is meant by "high-resolution defect."

A10: Thank you for your comments。"high-resolution defect" refers to the resolution of the defect in the imaging being high. The phrase appears in line 68: “Guided wave tomography (GWT), validated extensively since early 1990s [23-25], achieves high-resolution defect localization via meticulous mechanical scanning but demands dense sensor deployment due to its sensitivity, thus restricting widespread application owing to bulky expensive scanning equipment.”

The phrase "high-resolution defect" refers to the high resolution of defects in imaging. After considering your suggestion, I realized that this wording may cause misunderstanding, so I have changed it to "high defect resolution and defect location accuracy."

The corrections in the paper and the responds to the comments are described above, we have tried our best to improve the manuscript and made some changes in the manuscript, and hope the correction will meet with approval.

Once again, thank you for your comments and suggestions.

Round 2

Reviewer 1 Report

Comments and Suggestions for Authors

The authors have revised the manuscript according to the comments. 

Author Response

I am sincerely grateful for your acknowledgment of my manuscript. I sincerely hope that you enjoy good health and have a smooth career.

Reviewer 3 Report

Comments and Suggestions for Authors

The authors have improved the manuscript, but it still lacks detailed explanations of other damage imaging approaches. I will provide three examples to clarify the main concerns.

1) The authors mention: "While differing from the arrangement shown in the previous figure, the underlying principle of RAPID imaging remains unchanged." According to the RAPID imaging method, it is required to have pairs of sensors acting as actuators and receivers. The width of the ellipse is controlled by the scaling parameter beta, and the amplitudes by a damage index. However, since the receivers and actuators are on the same side, it is unclear how the elliptical distribution provides the images. Besides this critical question, when I asked for further explanations, I meant specific parameters. What is the scaling parameter beta? What is the damage index and why? These parameters significantly affect the results shown in Table 2. Estimated defect position errors of synthetic signals by imaging methods (mm).

2) Additionally, it is stated that: “Based on the above analysis, EVD imaging is actually based on DAS, where the input signal is processed with EVD to achieve noise reduction.” The problem is that readers cannot understand these explanations if authors do not provide details on what they mean by EVD imaging. Detailed explanations should be included in the manuscript, along with references to the EVD method and all the parameters that were considered.

3) The authors failed in providing further analysis of why each of these methods fails or succeeds. If no explanations are provided, the results in Table 2 suggest RAPID and EDV should not be used. 

Reproducibility is of key importance, and for that, the authors need to provide references, detailed parameters of the implemented approaches, and clear explanations so others can reproduce the results.

Comments on the Quality of English Language

No comment

Author Response

The specific reply can be found in the attached Word document.

Dear Editors and Reviewers:

Thank you for your letter and the comments concerning our manuscript entitled “A multi-strategy hybrid sparse reconstruction method based on spatial-temporal sparse wave number analysis for enhancing pipe ultrasonic guided wave anomaly imaging” (sensors-3117990). Those comments are all valuable and very helpful for revising and improving our paper, as well as the important guiding significance to our researches. We have studied comments carefully and have made correction which we hope meet with approval. Revised portion are marked in yellow in the paper. The corrections in the paper and the responds to the comments are as follows:

Q1: he authors mention: "While differing from the arrangement shown in the previous figure, the underlying principle of RAPID imaging remains unchanged." According to the RAPID imaging method, it is required to have pairs of sensors acting as actuators and receivers. The width of the ellipse is controlled by the scaling parameter beta, and the amplitudes by a damage index. However, since the receivers and actuators are on the same side, it is unclear how the elliptical distribution provides the images. Besides this critical question, when I asked for further explanations, I meant specific parameters. What is the scaling parameter beta? What is the damage index and why? These parameters significantly affect the results shown in Table 2. Estimated defect position errors of synthetic signals by imaging methods (mm).

A1: Thank you for your comments. I have concisely expounded the relevant theories of the ultrasonic guided wave imaging methods referred to in this manuscript within the revised version ranging from page 403 to 447, specifically as follows:

“Compare the proposed methods with conventional imaging methods, among which the expressions of the conventional imaging methods are as follows:

(1) Delay and sum (DAS) [26]

(7)

Where yi is the measured signal which is delayed dixy/cgc in the i-th wave path, dixy is the sum of the distances from the actuator to the point (x, y) and from the point (x, y) to the sensor, and cgc is the group velocity at the central frequency.

(2) Eigenvalue decomposition (EVD) [44]

Similar to DAS, prior to performing EVD imaging, delay processing of the measured signal is necessary. The delayed signal is designated as . The covariance matrix C of the delayed measured signal is resolved, and its eigenvalue decomposition is carried out as follows.

(8)

Where U is the matrix of eigenvectors, Σ is the eigenvalues corresponds to the main diagonal, λ is the eigenvalue of the defect scattering signal, and σ is the noise power. Based on the above eigenvalue decomposition, the largest eigenvalue is selected as the pixel value of the node (x, y).

(9)

(3) Reconstruction algorithm for probabilistic inspection of damage (RAPID) [28]

In RAPID, the baseline signal is needed to construct the signal difference coefficient (SDC). The expression of SDC is as follows.

(10)

Where Cdb is the covariance of detection signal and baseline signal, σd and σb are standard deviations of detection signal and baseline signal. The pixel value of the node (x, y) is expressed as:

(11)

(12)

(13)

(14)

Where β is the scaling parameter, and Ri(x,y) is the distance ratio between the indirect path and direct path linking the excitation and receiving sensors ,which is given in Formula 13. In i-th wave path, the location of excitation and receiving sensors are (xei,yei) and (xri,yri). The probability pi(x,y) is controlled by ti(x,y), ttof and td, where ti(x,y) denotes the sum of the actuator and receiver arrival times for (x, y) to the i-th path, , ttof is the arrival time of defect signal, and td denotes the defined TOF error.

(4) Multiple signal classification (MUSIC) [31]

The fundamental concept of the MUSIC algorithm, which is predicated on the uncorrelation between signal and noise, lies in decomposing the covariance matrix of array signals into the signal subspace and the noise subspace through conducting eigenvalue decomposition. Distinct from the aforementioned method, the positioning parameters of the MUSIC algorithm are (r, θ), and the conversion formulas between them and (x, y) are x = rcosθ, y = rsinθ. The covariance matrix C can be calculated by Formula 8, and it can be further be decomposed as:

(15)

Where the subscripts S and N represent the signal and noise subspaces, U and Σ are the eigenvectors and eigenvalues. Given Σ=diag[λ1, λ2,…, λL], and the eigenvalues can be arranged in descending order, λ1≥λ2≥…≥λk≥λk+1=…λL, where k is the number of the reflection source. So US and UN can be expressed as US=[e1,e2,…,ek] and UN=[ek+1,ek+2,…,eL], and the pixel value of the node (r, θ) can be calculated by:

(16)

Where A(r, θ)=[a1(r, θ), a2(r, θ),..., aL(r, θ)]T, and the expression of ai(r, θ) is as follows:

(17)

Where m is the serial number of the referential receiving sensor, d is the circumferential spacing of the receiving sensors. ”

Based on the aforesaid description of RAPID, the node pixel value is associated with the scaling parameter β and the signal difference coefficient (SDC). I conjecture that the scaling parameter beta and the damage index you referred to are β and SDC. Furthermore, although the excitation and receiving sensors in this paper are on the same side of the pipeline, the operational principle of RAPID remains unchanged. Generally, the excitation and receiving sensors of RAPID ought to be positioned at both ends of the defect, such that the defect lies within the ellipse with the excitation and receiving sensors as the foci, as depicted in the following figure, and the size of the elliptical area is governed by β. Evidently, the further the defect is from the excitation/receiving sensors, the smaller the pixel value.

In addition to the above-mentioned elliptical probability distribution, Reference 28 also proposed the ring probability distribution, which is shown as follows in the figure below. Using ToF as a parameter, an ellipse is constructed with the sensor position as the focal point where the damage may exist. The intersection point of the ellipse constructed by different paths is the location where the damage exists. However, since the error of sensor position paste, the error of wave speed, and the damage size will affect the accuracy of damage identification.

In Section 5.3, the axial distance between the excitation and reception is 30 mm, and the axial coordinates of defects D1 and D2 are 1000 mm and 2000 mm respectively. By comparing the above two probability distributions, it is obvious that the sensor setting and defect distribution in this paper are more suitable for the second one, that is, ring probability distribution. To ensure successful imaging by RAPID, it is essential to augment the elliptical area formed by the excitation and receiving sensors such that it can encompass defects D1 and D2. Consequently, a large value should be assigned to β, and 2×103 is selected in this paper. In addition, the parameter td is set as 1×10-4.

Finally, I would like to emphasize that in this paper, since the sensors are all set on the same side of the pipeline, which is different from the traditional RAPID sensor setting, the RAPID effect of this paper may be affected to some extent. However, because the theoretical dispersion curve used by RAPID, specifically the group velocity cgc at the central frequency, when the actual value of the dispersion curve differs significantly from the theoretical value, no matter how advanced the RAPID theory is, its positioning accuracy will inevitably be affected.

Q2: Additionally, it is stated that: “Based on the above analysis, EVD imaging is actually based on DAS, where the input signal is processed with EVD to achieve noise reduction.” The problem is that readers cannot understand these explanations if authors do not provide details on what they mean by EVD imaging. Detailed explanations should be included in the manuscript, along with references to the EVD method and all the parameters that were considered.

A2: Thank you for your comments. I have given the brief introduction of EVD, which is as follows:

“(2) Eigenvalue decomposition (EVD) [44]

Similar to DAS, prior to performing EVD imaging, delay processing of the measured signal is necessary. The delayed signal is designated as . The covariance matrix C of the delayed measured signal is resolved, and its eigenvalue decomposition is carried out as follows.

(8)

Where U is the matrix of eigenvectors, Σ is the eigenvalues corresponds to the main diagonal, λ is the eigenvalue of the defect scattering signal, and σ is the noise power. Based on the above eigenvalue decomposition, the largest eigenvalue is selected as the pixel value of the node (x, y).

(9)

Based on the abovementioned content, it is known that EVD imaging primarily acquires the maximum eigenvalue through operations such as signal delay, solving the covariance matrix, and eigenvalue decomposition, and uses it as the pixel value. None of the aforementioned operations involve parameter setting or control.

Q3: The authors failed in providing further analysis of why each of these methods fails or succeeds. If no explanations are provided, the results in Table 2 suggest RAPID and EDV should not be used. 

A3: Thank you for your comments. The result analysis regarding the imaging method is in lines 462 to 483, as follows:

“Due to the perturbation coefficient δ of 0.1 in the synthesized measurement signals, DAS, EVD, RAPID, and MUSIC based on the theoretical dispersion curve exhibits diminished accuracy in defect location. Especially, the axial defect errors of above methods are very large, varying from 111.5 mm to 180.5 mm. The circumferential defect error is only 1 mm for MUSIC, which is deemed satisfactory, whereas those of other methods range from 23 mm to 34 mm. The limitation in achieving high defect location accuracy using the above methods stems from the substantial disparity between the theoretical dispersion curve and its actual value, as well as the constrained number of sensors which hinders enhancements in location accuracy. In addition, as the measured signal contains both coherent and incoherent noise, the imaging method described above exhibits limited defect resolution. Therefore, threshold processing is essential to achieve satisfactory imaging results.

Compared to conventional UGW imaging methods, both MOMP and MHSR perform a search and optimization of dispersion parameters, resulting in dispersion parameter values that closely approximate the actual values. Consequently, the positioning accuracies of MOMPI and MHSRI are significantly enhanced. In particular, MHSR, which has pre-calculated dispersion curves using ST-SWA, possesses a more constrained and precise dispersion parameter search space compared to MOMP. Therefore, MHSRI shows a reduced defect location error of only (3 mm, 6.5 mm). Furthermore, as the input to MHSRI consists of the single-mode non-dispersive signals with dispersion compensation, both coherent and non-coherent noise are effectively eliminated. Consequently, only a limited number of pixel values exhibit non-zero in MHSRI imaging, indicating exceptionally high defect resolution.”

The corrections in the paper and the responds to the comments are described above, we have tried our best to improve the manuscript and made some changes in the manuscript, and hope the correction will meet with approval.

Once again, thank you for your comments and suggestions.
